# Clustered Influence Functions

**Miklós Máté Badó** [1]  **Kristian Fenech** [1]

## Abstract

Influence functions are a standard tool for data debugging and unlearning, but they become impractical for *high-query* subset workloads such as large-$K$ cross-validation, repeated resampling, or interactive what-if analysis as each subset query typically requires an expensive inverse-curvature solve. We introduce **Clustered Influence Functions (CiF)**, which turns subset influence into an *amortized subset oracle*. We build a compact cache once by clustering training gradients, solve a damped Generalised Gauss-Newton system only for cluster means, and answer new subset queries by a linear recombination using cluster membership counts. This yields per-query cost of $O(Cp)$ linear in the cache size $C$, and the number of model parameters $p$. We further provide a diagnostic error bound that decomposes approximation error into a *clustering scatter* term and a *solver residual* term, making the accuracy–compute trade-off explicit through the cache budget and solver tolerance. Evaluations across MNIST, CIFAR-10 show that CiF matches per-query influence rankings while significantly reducing the total runtime in high-$Q$ regimes, enabling influence-based workflows that are otherwise computationally prohibitive.

## 1. Introduction

Influence-style analyses ask *what would have changed* if parts of the training data were removed, reweighted, or corrected. These counterfactual "what-if" queries underpin practical workflows such as data debugging, unlearning triage, mislabel/poisoning detection, and large-$K$ cross-validation (CV). These queries are increasingly *subset-level* (users, sources, slices, time windows) and *high-query*: practitioners may evaluate hundreds to thousands of candi-

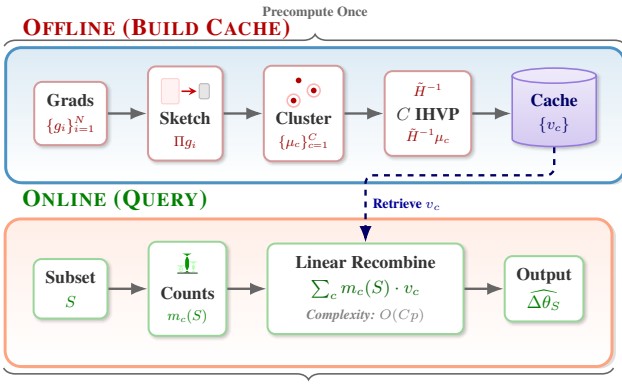

*Figure 1.* **CiF System Overview.** **Offline:** Gradients are sketched, clustered, and solved against the inverse curvature once. **Online:** For any subset query $S$, we simply count cluster memberships and linearly recombine cached vectors ($v_c$), bypassing expensive curvature solves entirely.

date subsets, often iteratively, before acting (Hammoudeh & Lowd, 2024; Wexler et al., 2020; Coalson et al., 2025; Ding et al., 2025; Epperson et al., 2025).

Classical influence functions estimate the effect of data removal via a first-order inverse-curvature approximation. Concretely, answering a subset query typically requires at least one expensive inverse Hessian-vector product (IHVP) solve. The cost scales with the number of queries. As a result, influence becomes infeasible in the regimes where it is most useful (e.g., large-$K$ CV, repeated resampling, or interactive what-if analysis over many candidate deletions).

This paper proposes that influence should be a reusable interface by leveraging the empirically observed low-rank structure of the deep gradient landscape (Zhang et al., 2025; Papyan et al., 2020; Mirzasoleiman et al., 2020; Charpiat et al., 2019; Fang et al., 2021). Consistent with theoretical work indicating that this structure supports ranking fidelity under Johnson-Lindenstrauss sketching (Hu et al., 2025).

We introduce **Clustered Influence Functions (CiF)**, an *amortized subset oracle* for influence under a fixed trained model and curvature proxy. Offline, CiF (i) sketches the gradients, to circumvent the cost of clustering in high-dimension $\mathbb{R}^p$ (ii) clusters the sketched gradients, and (iii) solves a damped generalised Gauss–Newton (GGN) sys-

[1]Department of Artificial Intelligence, Eötvös Loránd University, Budapest, Hungary. Correspondence to: Miklós Máté Badó <bado@inf.elte.hu>, Kristian Fenech <fenech@inf.elte.hu>.

*Proceedings of the 43rd International Conference on Machine Learning*, Seoul, South Korea. PMLR 306, 2026. Copyright 2026 by the author(s).

tem only for the *cluster-mean* gradients, storing the resulting IHVP responses as a cache. Online, a new subset query is answered by computing cluster membership counts and *linearly recombining* cached responses, yielding per-query cost that is small relative to a new IHVP solve. Figure 1 summarizes the offline/online process.

Beyond speed, CiF exposes a clean accuracy-compute trade-off. We provide an error bound that decomposes approximation error into (i) a *clustering scatter* term controlled by the cache/cluster budget and sketch quality, and (ii) a *solver residual* term controlled by the IHVP solver tolerance. This decomposition directly supports diagnostics: when CiF deviates from per-query influence, the bound identifies whether the bottleneck is representational (clusters too coarse) or numerical (solves too approximate).

We summarize our contributions as follows: (i) We formulate subset influence as a high-query workload and propose CiF, which decouples the cost of curvature inversion from the number of subset queries by caching cluster-level IHVP responses and answering new subsets by linear recombination. (ii) We prove an error decomposition that separates clustering-induced approximation from solver error, making the key trade-offs explicit through the cache budget and solver tolerance. (iii) On MNIST and CIFAR-10 we show that CiF closely matches per-query influence rankings for random subsets, tracks fold-wise CV risk over folds, and identifies mislabeled/poisoned subsets comparably to per-query influence while reducing marginal query cost substantially.

## 2. Related Work

Influence functions originate from robust statistics as a measure of local estimator sensitivity (Cook, 1977; Hampel, 1974) and were adapted to deep learning for data attribution and interpretability (Koh & Liang, 2017; Koh et al., 2019). While theoretically elegant, their application to large, non-convex models is fragile (Basu et al., 2021), motivating stabilized formulations such as proximal influence analysis (Bae et al., 2022) and second-order group extensions (Basu et al., 2020). In practice, influence estimation relies on matrix-free curvature products (Pearlmutter, 1994; Schraudolph, 2002; Martens, 2010; Botev et al., 2017) and regularized solvers (Tikhonov & Arsenin, 1977), but the per-query inverse-Hessian-vector product (IHVP) cost remains a bottleneck. Subsequent efforts explored scalable approximations using low-rank curvature structures and stochastic solvers (Schioppa et al., 2022; Koh et al., 2019; Klochkov & Liu, 2024).

Recent literature explores efficiency by relaxing the influence objective itself. FastIF and TracIn approximate influence by restricting evaluation to local neighborhoods

or first-order gradient trajectories, effectively altering the underlying attribution metric for speed (Guo et al., 2021; Pruthi et al., 2020). Similarly, projection-based paradigms like TRAK reframe attribution as a kernel regression problem on random features, trading exact curvature fidelity for constant-time inference (Park et al., 2023). These developments leverage the same low-rank gradient geometry utilized in coreset selection (Zhang et al., 2025; Sridharan et al., 2025; Papyan et al., 2020; Fang et al., 2021; Mirzasoleiman et al., 2020; Killamsetty et al., 2021); however, a critical distinction remains: while coreset methods optimize data selection for *forward training* and TRAK approximates a *linearized proxy*, our work retains the rigorous Inverse-Hessian formulation of standard influence. CiF is designed to accelerate the *exact* curvature-based diagnostic without substituting it for a heuristic proxy, positioning it as an amortized accelerator for the standard influence definition rather than a redefined metric.

Clustered Influence Functions achieves this by targeting the operand redundancy of the influence equation directly, rather than relaxing the curvature operator itself. Instead of altering the metric, we compress the input space by subjecting high-dimensional gradients to a Johnson-Lindenstrauss transform (Johnson et al., 1984; Cohen et al., 2015; Makarychev et al., 2019), we enable scalable clustering that preserves the pairwise geometry required for Hessian inversion. This formulation allows us to explicitly decompose the approximation error into a *geometric term* (governed by within-cluster scatter) and a *numerical term* (governed by solver residual), thereby creating a link between computational amortization and the statistical phenomenon of neural collapse (Papyan et al., 2020; Fang et al., 2021).

## 3. Methodology

### 3.1. Problem Formulation: Amortized Subset Influence

We address the problem of *subset influence* estimation under high-query workloads. Let $\mathcal{D} = \{(x_i, y_i)\}_{i=1}^N$ be the training set and $\theta_0 \in \mathbb{R}^p$ be parameters minimizing the regularized empirical risk $R(\theta) = \frac{1}{N} \sum_{i=1}^N L_i(\theta) + \lambda_{wd} \|\theta\|_2^2$. Our goal is to serve a workload of $Q$ distinct subset removal queries, $\mathcal{S} = \{S_1, \ldots, S_Q\}$, where each $S_k \subset \{1, \ldots, N\}$. For any query $S$, we seek to approximate the parameter change $\Delta\theta_S$ and the resulting loss change on an evaluation set $V$, $\Delta\ell(S; V)$, without incurring the cost of a fresh inverse-curvature product for each query. We use *fold query* when $S$ is one fold from a cross-validation partition, and *leave-group-out query* when $S$ is one of many small predefined groups.

Classical approaches require $\mathcal{O}(Q \cdot N_{\text{steps}})$ curvature-vector products. In contrast, we propose an *amortized* framework.

We pay a one-time offline cost to construct a compact *influence cache*, after which we resolve arbitrary subset queries via lightweight linear recombination.

## 3.2. Preliminaries

We adopt the standard Generalised Gauss–Newton (GGN) (Martens, 2020) approximation for curvature, which is positive semi-definite. The GGN matrix is defined as

$$H_{\text{GN}} = \frac{1}{N} \sum_{i=1}^{N} J_i^\top W_i J_i, \tag{1}$$

where $J_i = \nabla_\theta f_\theta(x_i)|_{\theta_0}$ is the Jacobian of the model output, and $W_i = \nabla_{\hat{y}}^2 \ell(\hat{y}, y_i)$ is the Hessian of the loss w.r.t. the model output. To ensure invertibility and numerical stability, we employ Tikhonov damping (Tikhonov & Arsenin, 1977):

$$\tilde{H} := H_{\text{GN}} + \lambda I. \tag{2}$$

Under the standard linear influence approximation, the removal of a subset $S$ induces a parameter update:

$$\Delta\theta_S \approx \frac{1}{N} \tilde{H}^{-1} \sum_{i \in S} \nabla_\theta L_i(\theta_0). \tag{3}$$

The computational bottleneck lies in evaluating $\tilde{H}^{-1}v$ for the distinct gradient sum $v = \sum_{i \in S} g_i$ of each query, with $g_i$ as the per-sample gradient of the $i$-th datapoint. We circumvent this by exploiting the low-rank structure of the gradient space.

Classical influence functions treat a trained estimator as a functional of the data distribution and characterize local sensitivity via first-order derivatives (influence curves) (Hampel, 1974). A common modern instantiation considers an upweighted objective $R_\epsilon(\theta) := R(\theta) + \epsilon L_i(\theta)$ with minimizer $\theta_\epsilon$. Under differentiability and local invertibility of the stationarity conditions around $\theta_0$, the implicit function theorem yields (Koh & Liang, 2017)

$$\left. \frac{d\theta_\epsilon}{d\epsilon} \right|_{\epsilon=0} = -H^{-1} \nabla_\theta L_i(\theta_0), \tag{4}$$

where $H = \nabla_\theta^2 R(\theta_0)$. For subset removal, the same linearization implies a group update proportional to $H^{-1} \sum_{i \in S} g_i$. Influence is fundamentally local, and empirically can degrade in deep, nonconvex models and for larger perturbations, motivating stabilized curvature proxies and amortized approximations (Basu et al., 2021; Koh et al., 2019).

Every subset query $S$ enters only through the gradient sum $\sum_{i \in S} g_i$. If per-example gradients can be partitioned into clusters with small within-cluster variation, then $\sum_{i \in S} g_i$

can be approximated by a linear combination of cluster representatives. This is conceptually aligned with gradient-matching subset selection/coreset methods that approximate full gradients using a small weighted subset in gradient space (Mirzasoleiman et al., 2020; Killamsetty et al., 2021).

Clustering in $\mathbb{R}^p$ is often impractical, so we use a Johnson–Lindenstrauss (JL) sketch $\Pi \in \mathbb{R}^{d_\Pi \times p}$ to cluster in $\mathbb{R}^{d_\Pi}$, where random projections preserve pairwise distances up to $(1 \pm \varepsilon)$ with high probability for $d_\Pi = O(\varepsilon^{-2} \log N)$ (Johnson et al., 1984). We emphasize that the sketch affects only the *partition*; cluster means and all curvature solves are computed in the original parameter space.

## 3.3. Subset-agnostic influence cache

Computing equation 3 for many queries requires a new solve for each distinct right-hand side $\sum_{i \in S} g_i$, which becomes prohibitive when the number of queries $Q$ is large. We therefore build a reusable cache by clustering gradients and solving only $C$ systems offline.

**Phase I: sketching and clustering gradients (offline)**
To avoid storing and clustering full gradients, we first compute the JL sketch

$$h_i = \Pi g_i, \qquad \Pi \in \mathbb{R}^{d_\Pi \times p}, \tag{5}$$

using a streaming manner (detailed in Appendix C) and cluster $\{h_i\}_{i=1}^N$ into $C$ clusters (e.g., $k$-means in $\mathbb{R}^{d_\Pi}$):

$$\{h_i\}_{i=1}^N \xrightarrow{k\text{-means}} C \text{ clusters.} \tag{6}$$

Let $c(i) \in \{1, \ldots, C\}$ be the assignment and $n_c$ cluster sizes. We then compute the full-space mean gradient per cluster

$$\mu_c = \frac{1}{n_c} \sum_{i:c(i)=c} g_i. \tag{7}$$

**Phase II: $C$ damped GGN solves (offline)** For each cluster $c$, we solve one linear system

$$\tilde{H} v_c = \mu_c, \tag{8}$$

using CG with matrix-vector products. We stop when the relative residual satisfies

$$\rho_c := \frac{\|\mu_c - \tilde{H} v_c\|_2}{\|\mu_c\|_2} \leq \rho_{\text{tol}}. \tag{9}$$

We cache $\{v_c\}_{c=1}^C$.

**Phase III: answering subset queries (online)** Given a query subset $S$, compute cluster counts $m_c(S) := |\{i \in S : c(i) = c\}|$ and form the cached parameter-update estimator

$$\hat{\Delta}\theta_S := \frac{1}{N} \sum_{c=1}^C m_c(S) v_c. \tag{10}$$

For loss prediction on evaluation set $V$, let $\alpha_c(V) := g_V^\top v_c$ and compute,

$$\Delta\hat{\ell}(S;V) := g_V^\top \hat{\Delta}\theta_S = \frac{1}{N}\sum_{c=1}^{C} m_c(S)\,\alpha_c(V) \quad (11)$$

When the evaluation set $V$ is fixed across many queries (e.g., a single validation set), the scalars $\{\alpha_c(V)\}_{c=1}^{C}$ can be precomputed once, after which each query reduces to counting and an $O(C)$ recombination in equation 11. For fold queries where $V$ varies with $k$, one can precompute the $K \times C$ matrix of dot-products $\alpha_c(S_k)$ to enable $O(C)$ online evaluation per fold.

### 3.4. Complexity and storage

**Offline cost.** Cache construction decouples curvature inversion from the query count $Q$. The fixed offline cost comprises gradient computation ($\mathcal{O}(Np)$), JL sketching ($\mathcal{O}(Npd_\Pi)$), clustering ($\mathcal{O}(NCd_\Pi I)$), and solving $C$ inverse-curvature systems ($\mathcal{O}(CN_{CG}Np)$). This investment amortizes heavily over the workload. While standard influence scales linearly as $T_{\text{IF}} \approx Q \cdot t_{\text{IHVP}}$, our method scales as $T_{\text{CiF}} \approx C \cdot t_{\text{IHVP}} + Q \cdot t_{\text{lin}}$, becoming strictly faster once the query volume satisfies $Q > C$.

**Online cost.** Online resolution is near-instantaneous. For a query $S$, counting memberships costs $\mathcal{O}(|S|)$. Reconstructing the full parameter update requires $\mathcal{O}(Cp)$ linear recombination operations. However, for loss prediction on fixed evaluation sets (e.g., cross-validation folds), we pre-calculate projection scalars $\alpha_c(V) \triangleq g_V^\top v_c$ offline. This optimization reduces the online cost to a negligible $\mathcal{O}(|S| + C)$ scalar sum, bypassing $\mathcal{O}(Cp)$ vector operations entirely. For dynamic evaluation sets where $V$ is not known offline, we incur an additional $\mathcal{O}(Cp)$ dot-product cost per target to project the cached vectors.

**Storage.** The memory footprint is dominated by the influence cache $\{v_c\}_{c=1}^{C} \subset \mathbb{R}^p$, requiring $\mathcal{O}(Cp)$ floats. Auxiliary storage for cluster assignments $\{c(i)\}_{i=1}^{N}$ and sketches $\{h_i\}_{i=1}^{N}$ is negligible. While this exceeds the $\mathcal{O}(p)$ storage of standard IF, it trades memory for the ability to serve effectively infinite queries without re-computation.

### 3.5. Error bounds: deterministic control and expected-case corollary

We separate approximation error into (i) a *clustering* term capturing how well cluster means represent individual gradients and (ii) a *solver* term capturing CG residuals. The formal proof is in Appendix A.

Let $r_c := \mu_c - \tilde{H}v_c$ denote the cached linear-system residual for cluster $c$. For any subset $S$,

$$\Delta\theta_S - \hat{\Delta}\theta_S = \frac{1}{N}\tilde{H}^{-1}\left(\sum_{i\in S} g_i - \sum_{c=1}^{C} m_c(S)\mu_c\right) \\ + \frac{1}{N}\tilde{H}^{-1}\left(\sum_{c=1}^{C} m_c(S)r_c\right), \quad (12)$$

hence using $\|\tilde{H}^{-1}\|_2 \leq 1/\lambda$ we obtain

$$\|\Delta\theta_S - \hat{\Delta}\theta_S\|_2 \leq \frac{1}{\lambda N}\left\|\sum_{i\in S}\left(g_i - \mu_{c(i)}\right)\right\|_2 \\ + \frac{1}{\lambda N}\sum_{c=1}^{C} m_c(S)\,\|r_c\|_2. \quad (13)$$

Moreover, for any evaluation set $V$,

$$\left|\Delta\hat{\ell}(S;V) - \widehat{\Delta\ell}(S;V)\right| \leq \|g_V\|_2\,\|\Delta\theta_S - \hat{\Delta}\theta_S\|_2, \quad (14)$$

linking parameter-error control directly to loss-delta accuracy.

Define the within-cluster scatter

$$\text{tr}(\bar{\Sigma}) := \frac{1}{N}\sum_{i=1}^{N}\|g_i - \mu_{c(i)}\|_2^2. \quad (15)$$

When $S$ is drawn uniformly among all subsets of size $m$ (a standard finite-population sampling model), the first term in equation 13 admits an explicit expected bound in terms of $\text{tr}(\bar{\Sigma})$, yielding the following corollary.

**Theorem 3.1** (Expected parameter error against per-query influence). *Assume $\tilde{H} \succeq \lambda I_p$ with $\lambda > 0$. Let $S$ be drawn uniformly among all subsets of $\{1,\ldots,N\}$ of size $m$. Then*

$$\mathbb{E}\|\Delta\theta_S - \hat{\Delta}\theta_S\|_2 \leq \frac{1}{\lambda N}\sqrt{\frac{m(N-m)}{N-1}\,\text{tr}(\bar{\Sigma})} \\ + \frac{m}{\lambda N}\sum_{c=1}^{C}\frac{n_c}{N}\,\rho_c\,\|\mu_c\|_2, \quad (16)$$

*where $\rho_c$ is the relative residual in equation 9.*

The first term decreases as clusters tighten (smaller $\text{tr}(\bar{\Sigma})$), while the second decreases as cached solves become more accurate (smaller $\rho_c$). Both terms improve with stronger damping (larger $\lambda$) and degrade with larger subset sizes $m$ (for $m \leq N/2$), matching the intuition that larger edits amplify both clustering mismatch and solver error.

## 4. Experiments

### 4.1. Experimental Protocol

We evaluate on MNIST (LeNet-5 (Lecun et al., 1998)) and CIFAR-10 (Krizhevsky et al., 2009) (AlexNet (Krizhevsky

et al., 2012), ResNet-18 (He et al., 2016)). For MNIST, we additionally employ a stratified 10% subsample (MNIST-10%) to facilitate expensive exact Leave-One-Out (LOO) comparisons. We additionally report results on ImageNet-1k (Deng et al., 2009) using a DeiT-Tiny (Touvron et al., 2021) pretrained model.

We follow the CNN training protocol of Bae et al. (2022). All models are trained using SGD with momentum 0.9, weight decay $5 \times 10^{-4}$, and batch size 128 for 200 epochs. Base learning rates are selected via validation accuracy sweeps over $\{1.0, 0.3, \ldots, 0.001\}$. We use a fixed learning rate for MNIST and a step-decay schedule (factor of 5 at epochs $\{60, 120, 160\}$) for CIFAR-10. All influence estimators operate on a *fixed* model snapshot $\theta_0$.

## 4.2. Baselines and Implementation

We compare **CiF** against a suite of standard and state-of-the-art attribution methods: (i) **Exact LOO:** Cold-start retraining, (ii) **Per-query IF:** Standard Influence Functions using a damped Gauss–Newton (GGN) curvature proxy, (iii) **PBRF:** Proximal Bregman Response Functions (Bae et al., 2022), (iv) **Approximations:** First-order gradient similarity (**Grad-Dot** (Deng et al., 2024; Charpiat et al., 2019)), **TRAK** (Park et al., 2023).

**Hyperparameters & Solvers.** For all curvature-based methods (CiF, IF, PBRF), we use a Tikhonov damping of $\lambda = 10^{-3}$ ($10^{-1}$ for DeiT-Tiny).

- **IF Baseline:** For each query subset $S$, we compute the gradient sum and solve the inverse-curvature system using Conjugate Gradients (CG). We use a relative tolerance of $10^{-7}$ ($10^{-3}$ for ImageNet).

- **CiF (Ours):** We project per-example gradients using a dense Gaussian Johnson–Lindenstrauss sketch and cluster the projected features via $k$-means++ into $C$ clusters. We solve the damped GGN system *once* per cluster mean (using the same $\lambda$ and CG settings as IF) to build the cache. Query responses are synthesized via linear recombination of cached solutions.

## 4.3. Evaluation Tasks

We evaluate CiF along three dimensions: **(i) Robustness.** Starting from a broad ablation over cache hyperparameters $(C, d_{\Pi})$ on MNIST-10%/LeNet (full grids in Appendix B.1), we select representative settings and compare fold-level $\Delta$loss rankings against leave-fold-out retraining (LOO) and PBRF (Bae et al., 2022) via correlation matrices (Fig. 2). **(ii) Scaling and reuse.** On CIFAR-10 across architectures, we reuse a single cache across $K \in \{50, 100\}$ fold workloads and report end-to-end time,

speedups, and fold-level agreement with IF with 95% bootstrap CIs (Table 1). **(iii) Amortization regimes.** We sweep cache size $C$ and query count $Q$ to measure when the one-time cache build is repaid, yielding a phase diagram and break-even frontier (Fig. 3). We additionally include an ImageNet-1k/DeiT-Tiny stress test under high-$K$ leave-group-out ($K=5000$) using a truncated query protocol reporting the speed/accuracy trade-off for $C \in \{128, 512\}$ (Table 2). We additionally simulate a data debugging scenario by poisoning a random subset of $G = 500$ groups (training indices only). Poisoned examples are relabeled to a target class and stamped with a square patch trigger. We evaluate robustness by varying the trigger from *fixed* (structured signal) to *random* locations (noisy signal) (Fig. 4).

We report Spearman and Pearson correlations for ranking consistency and CV loss approximation. For poison detection, we report PR-AUC and Recall@$k$. Uncertainty is estimated via bootstrap percentile intervals ($B = 10,000$) or averages over three random seeds $\{0, 10, 100\}$. In Appendix D we report runtime and memory usage breakdowns.

# 5. Results

## 5.1. Robustness

We first ablate the cache hyperparameters (cluster budget $C$ and JL sketch dimension $d_{\Pi}$) across multiple $K$ on MNIST-10% with LeNet, the full grids are reported in the appendix B.1. From these ablations we select six representative configurations (two best, two worst, and two 'average' by fold-level agreement) and then rerun them against (i) LOO and (ii) PBRF. Figure 2 shows one representative setting ($K=500$, $C=10$); the remaining selected settings follow the same evaluation protocol and are included in the appendix. We report rank agreement across folds (Spearman/Pearson), so the key readout is whether methods induce similar fold orderings rather than matching absolute scales.

The main pattern in Figure 2 is that the closest pairwise agreement occurs within the IF-style family (IF vs. CiF) and between IF/CiF and PBRF, while comparisons involving LOO are systematically lower. In particular, CiF tracks IF more closely than it tracks LOO, and PBRF is the nearest comparator to IF/CiF among the baselines in this experiment. This supports using CiF as a drop-in surrogate for IF when the goal is to preserve fold-level rankings implied by the IF objective; the larger gap relative to LOO is shared by IF and CiF rather than being specific to caching.

## 5.2. CiF vs IF on CIFAR-10

Table 1 evaluates cache reuse across fold workloads by treating $K \in \{50, 100\}$ as a single run: IF pays for 150 fold

*Table 1.* **CiF cache is reusable across fold workloads on CIFAR-10.** We treat evaluating $K \in \{50, 100\}$ as a single workload: IF time is the sum of the two runs (150 fold queries total), while CiF pays the cache-build cost once per $C$ and reuses it to answer both $K{=}50$ and $K{=}100$ fold queries. Agreement is computed over folds for each $K$; we report point estimates with 95% bootstrap percentile CI.

| MODEL | IF TIME (H) ($K = 50{+}100$) | $C$ | CiF TIME (H) | SPEED UP (IF/CiF) | PEARSON (95% CI) $K{=}50$ | PEARSON (95% CI) $K{=}100$ | SPEARMAN (95% CI) $K{=}50$ | SPEARMAN (95% CI) $K{=}100$ |
|---|---|---|---|---|---|---|---|---|
| LENET | 32.1 | 10 | 2.1 | **15.3×** | 0.943 [0.890,0.974] | 0.939 [0.900,0.964] | 0.934 [0.850,0.973] | 0.940 [0.900,0.963] |
|  |  | 50 | 10.7 | **3.0×** | 0.942 [0.886,0.974] | 0.938 [0.898,0.964] | 0.934 [0.852,0.973] | 0.940 [0.899,0.963] |
| ALEXNET | 63.3 | 10 | 5.4 | **11.7×** | 0.994 [0.988,0.996] | 0.942 [0.920,0.973] | 0.987 [0.964,0.993] | 0.992 [0.981,0.995] |
|  |  | 50 | 22.2 | **2.9×** | 0.994 [0.989,0.996] | 0.958 [0.925,0.991] | 0.987 [0.965,0.994] | 0.988 [0.970,0.994] |
| RESNET18 | 75.0 | 10 | 5.4 | **13.9×** | 0.740 [0.591,0.845] | 0.761 [0.678,0.827] | 0.746 [0.554,0.866] | 0.771 [0.663,0.846] |
|  |  | 50 | 25.4 | **3.0×** | 0.935 [0.897,0.966] | 0.926 [0.902,0.957] | 0.946 [0.888,0.970] | 0.941 [0.901,0.963] |

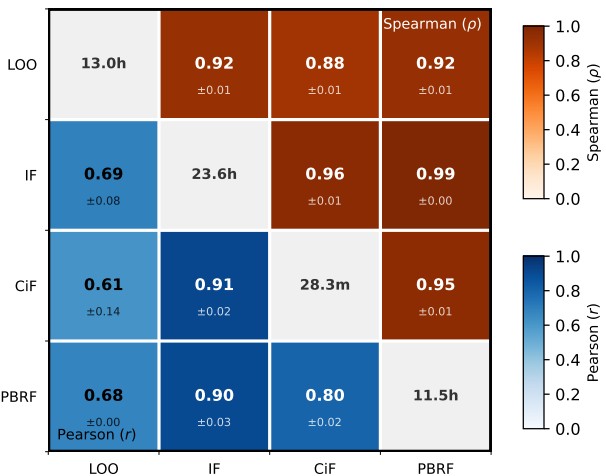

*Figure 2.* **Correlation matrix for fold-level $\Delta$loss predictions (MNIST-10% LeNet; $K{=}500$, $C{=}10$, $d_\Pi{=}32$).** Rows/columns compare leave-one-out retraining (LOO), influence functions (IF), our cached clustered estimator (CiF), and the proximal Bregman response function (PBRF). Each off-diagonal cell shows *upper*: Spearman $\rho$ and *lower*: Pearson $\rho$ over the $K$ folds; color encodes correlation. Diagonal entries report end-to-end runtimes $T$.

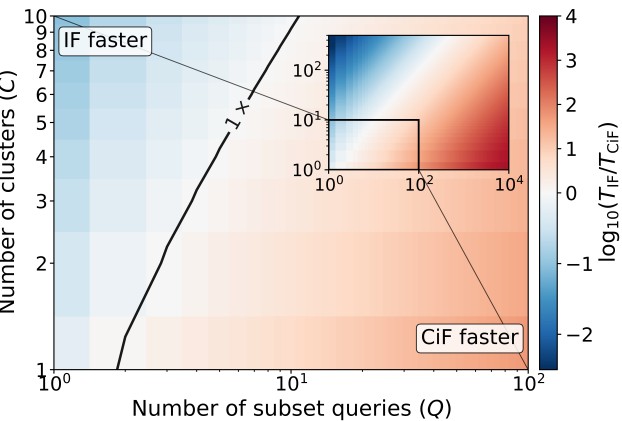

*Figure 3.* **CiF vs IF amortization phase diagram (CIFAR-10 AlexNet).** The color encodes speedup of CiF over IF as a function of cluster budget $C$ and number of queries $Q$. The dashed line shows the theoretical break-even frontier. The inset highlights that cache reuse across multiple $K$-fold runs (or other subset queries) pushes the operating point to larger $Q$ without rebuilding, deepening the speedup.

queries in total (50+100), whereas CiF builds a cache once per cluster budget $C$ and reuses it to answer both $K$ settings. For LeNet and AlexNet, this reuse yields large amortized gains at $C{=}10$ (15.3× and 11.7× speedup, respectively) while maintaining high agreement with IF across both $K$ values (Spearman 0.93–0.99). Increasing the cache size to $C{=}50$ reduces the speedup to $\sim 3\times$ but leaves agreement largely unchanged, indicating that most of the benefit comes from avoiding repeated solves across folds. For ResNet18, the $K{=}50$ results show that small caches can underfit the fold-level variation ($C{=}10$ yields substantially lower correlation than $C{=}50$), supporting that more complex models need larger caches for accurate approximation.

## 5.3. Amortization and cache reuse

Figure 3 characterizes the efficiency gap between standard IF and our clustered cache. Standard IF scales linearly with the number of queries $Q$, costing $T_{\mathrm{IF}} \approx Q \cdot t_{\mathrm{IHVP}}$. In contrast, CiF pays a fixed upfront cost for $C$ cluster solves plus a negligible recombination cost per query: $T_{\mathrm{CiF}} \approx C \cdot t_{\mathrm{IHVP}} + t_{\mathrm{over}} + Q \cdot t_{\mathrm{lin}}$.

The break-even frontier occurs at $Q \approx C + (t_{\mathrm{over}}/t_{\mathrm{IHVP}}) \approx C$. Thus, CiF becomes strictly faster as soon as the query count exceeds the cluster budget. Once the cache is built, the marginal cost of additional queries vanishes, allowing extensive downstream tasks-such as re-running CV with different splits or auditing thousands of potential poison candidates.

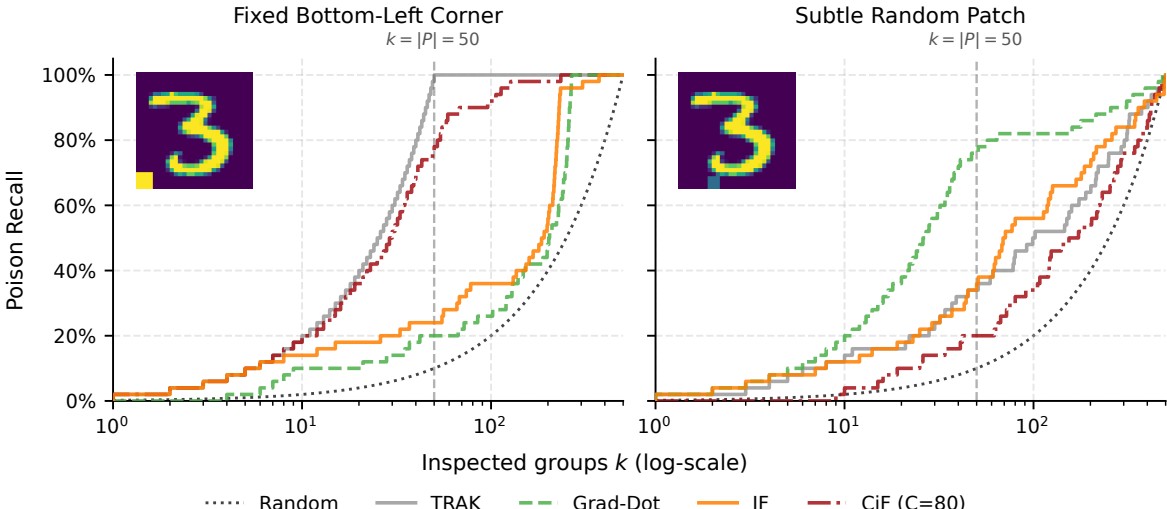

*Figure 4.* **Backdoor group retrieval on MNIST.** Poison recall versus inspected groups $k$ (log-scale); the dashed line marks the true number of poisoned groups ($|\mathcal{P}| = 50$). **Left:** Fixed Bottom-Left trigger. A weak but directionally consistent poison signal enables CiF to reduce score variance via aggregation, yielding higher early recall than per-example influence baselines. **Right:** Random-location trigger. The absence of a shared trigger direction causes aggregation to cancel the signal, and per-example methods outperform CiF.

## 5.4. CiF vs IF on ImageNet

*Table 2.* **ImageNet-1k (DeiT-Tiny).** High-$K$ ($K = 5000$) leave-group-out analysis. Times for IF are projected from the $K_{\text{eval}} = 50$ subsample.

| METHOD | BUILD (1×) | QUERY ($S_k$) | TOTAL (FULL $K$) | SPEEDUP | RANK $\rho$ |
|---|---|---|---|---|---|
| IF | – | 7.5 M | $\approx 625$ H | 1.0× | – |
| **CiF** ($C$=512) | 67 H | < 0.1 S | 67 H | 9.3× | 0.19 |
| CiF ($C$=128) | 19 H | < 0.1 S | 19 H | 32.9× | 0.07 |

We conduct a stress-test on ImageNet-1k using a pretrained DeiT-Tiny model ($N \approx 1.28$M) under a high-cardinality leave-group-out workload with $K = 5000$ micro-folds (approximately 250–300 samples each). For tractability, we use a truncated evaluation protocol: we uniformly sample $K_{\text{eval}} = 50$ query folds. For each query fold $v$, we compute a single inverse-curvature solve $s_v = (\widehat{G} + \lambda I)^{-1} g_v$ using CG, and score all target folds $t \neq v$ via dot products $\text{IF}(v \to t) = -g_t^\top s_v$, yielding a full ranking over all $K - 1$ targets while requiring only one linear solve per evaluated query fold.

To further control cost at ImageNet scale, we construct the curvature proxy $\widehat{G}$ from a subset of 10,000 uniformly sampled training examples (instead of the full $N \approx 1.28$M) and hold this proxy fixed across all queries. This corresponds to a standard subsampled second-order approximation, where the Newton system is solved with respect to an approximate curvature operator rather than the full-dataset Hessian/Gauss–Newton matrix (Ye et al., 2021). CiF is evaluated on the same query folds and targets: for each target fold $t$, CiF produces an approximate parameter shift

$\Delta \theta_t \approx -(\widehat{G} + \lambda I)^{-1} \sum_{i \in t} g_i$ from a single precomputed cache, and scores are computed as $\text{CiF}(v \to t) = g_v^\top \Delta \theta_t$.

We report Spearman rank correlation between CiF and IF scores, averaged over the $K_{\text{eval}}$ query folds. All ImageNet runs use JL sketch dimension $d_\Pi = 256$, and we report cache sizes $C \in \{512, 128\}$ (Table 2). At ImageNet scale with $C \in \{128, 512\}$, the cache compresses approximately 1.28M examples into very coarse partitions of a highly heterogeneous training set, so within-cluster scatter remains large and the induced approximation error produces rank inversions. This is consistent with the observed monotone improvement when increasing cache size ($\rho$ rises from 0.07 at $C$=128 to 0.19 at $C$=512), indicating that additional clusters translate into better agreement with IF. Furthermore, this result is aligned with Theorem 3.1.

## 5.5. CiF amortization with a projected influence backend

We also test whether CiF composes with a modern projected influence backend rather than depending on CG. On CIFAR-10/ResNet-18, we use a LoGra-style PCA/KFAC backend (Choe et al., 2026) to populate the influence cache and compare the online query cost of LoGra against CiF+LoGra. Offline artifact construction is excluded for both methods, so this experiment isolates the marginal cost of serving repeated subset queries after the relevant backend artifacts have been built. Across $K \in \{50, 100, 500, 1000\}$ and $C \in \{32, 128, 512\}$, corresponding to 4950 query evaluations over 12 $(K, C)$ configurations, CiF+LoGra reduces mean online query time from 8.68 ms to 3.45 ms, a 2.51× improvement (Table 3). At the

*Table 3.* **Online query timing for LoGra and CiF+LoGra on CIFAR-10/ResNet-18.** Offline artifact construction is excluded for both methods. Results are aggregated over $K \in \{50, 100, 500, 1000\}$ and $C \in \{32, 128, 512\}$, giving 4950 query evaluations across 12 configurations. Correlations report agreement with per-query IF, averaged across configurations.

| METHOD | TOTAL ONLINE TIME | QUERIES | MEAN/QUERY |
|---|---|---|---|
| LoGra | 42.95 s | 4950 | 8.68 ms |
| CiF+LoGra | 17.09 s | 4950 | 3.45 ms |
| SPEEDUP | 2.51× ONLINE QUERY SPEEDUP | | |

same time, CiF+LoGra maintains strong agreement with per-query IF, with mean Pearson $0.879 \pm 0.063$ and Spearman $0.875 \pm 0.039$ across configurations. This supports the intended interpretation of CiF as a workload-level amortization layer: the cache-and-recombine mechanism is not tied to a particular IHVP solver, and projected solvers can be used inside cache construction to reduce the cost of larger cache budgets.

### 5.6. Backdoor Poisoning Detection

We study backdoor detection on a controlled 10% stratified subset of MNIST ($N = 6,000$), partitioned into $G = 500$ disjoint training groups. A subset $\mathcal{P}$ of $|\mathcal{P}| = 50$ groups (10%) is poisoned by inserting a patch trigger and relabeling to a fixed target class ($y = 0$). We consider two regimes: (i) **Fixed Location**, where the trigger is placed consistently at the bottom-left of poisoned examples (Attack Success Rate (ASR) 98% on a triggered test set), and (ii) **Random Location**, where a low-contrast trigger is placed at random coordinates per example (ASR 12%), serving as a sanity-check with weak trigger dependence.

Poison detection is inherently a *ranking* problem rather than a pointwise estimation task. Practitioners inspect the top-$k$ most influential groups, and success is measured by Recall@k or PR-AUC rather than numerical accuracy. Prior work shows that influence estimators optimized for unbiasedness can perform poorly under ranking metrics due to high variance and magnitude bias, where clean but difficult samples dominate rankings despite lacking causal relevance (Barshan et al., 2020; Hammoudeh & Lowd, 2024).

This failure mode is amplified in deep networks. Basu et al. (Basu et al., 2021) demonstrate that per-example influence functions are highly unstable with respect to initialization, model capacity, and damping, while Hammoudeh and Lowd (Hammoudeh & Lowd, 2024) identify low signal-to-noise ratio (SNR) and gradient cancellation as central limitations of dot-product-based attribution. The spectral structure of deep-model Hessians further amplifies sample-specific noise under inversion, producing heavy-tailed, high-variance influence scores that are ill-suited for

reliable ranking (Koh et al., 2019; Feldman & Zhang, 2020; Gurbuzbalaban et al., 2021).

To interpret when aggregation improves detection, we use the decomposition

$$g_i = \alpha_i v_{\text{signal}} + \varepsilon_i,$$

where $v_{\text{signal}}$ denotes a trigger-aligned direction shared across poisoned samples, $\alpha_i$ its strength, and $\varepsilon_i$ high-dimensional sample-specific noise. Empirical analyses of gradient geometry show that such noise is largely orthogonal to task-relevant or backdoor-related directions and dominates gradient variance in overparameterized models (Hölzl et al., 2025). In the **Fixed Location** regime, poisoned examples exhibit weak but directionally consistent alignment with $v_{\text{signal}}$, while within-group cosine similarity remains near zero. Clustering and averaging therefore act as geometric regularization: they suppress orthogonal noise while preserving the shared low-rank signal, yielding lower-variance rankings despite approximation bias. This aligns with prior observations that poisoned samples form coherent spectral signatures in representation or gradient space (Tran et al., 2018; Chen et al., 2018).

In contrast, in the **Random Location** regime, trigger placement varies across examples and no consistent trigger-aligned direction exists. Aggregation cancels both noise and signal, degrading ranking performance, and per-example-resolution methods are better matched to the data geometry. Overall, these regimes support a variance-centric view of poisoning detection: CiF improves retrieval when poisoning induces a weak but systematically shared gradient component, and degrades when the learned signal is highly dispersed. This behavior is consistent with Theorem 3.1, which controls approximation bias to IF, while detection performance is governed by variance reduction and SNR (Barshan et al., 2020).

## 6. Discussion

CiF reframes influence estimation as an *amortized subset oracle*. Instead of solving an inverse-curvature system per subset, we build a compact cache once (under a fixed trained model and curvature proxy) and reuse it across many subset queries. This makes *high-query* influence workloads practical, e.g., large-$K$ cross-validation, repeated resampling, and interactive what-if analyses, while preserving the same damped-GGN influence semantics as per-query IF baselines (Koh & Liang, 2017; Koh et al., 2019).

CiF should be used when many subset queries will be served under the same trained model and curvature proxy, when $Q \gg C$, and when the within-cluster scatter diagnostic decreases sufficiently at feasible cache budgets. CiF

is not suitable for one-off queries, rapidly changing models, exact retraining estimation, or regimes where the cache budget is too small to represent gradient heterogeneity. In such cases, the clustering term in Theorem 3.1 remains large and the cached approximation should be expected to have low rank fidelity.

The main scaling bottleneck is the $O(Cp)$ cache. Promising directions to improve this include layer-wise caches, low-rank or subspace caches storing $O(Cr)$ coefficients, quantized cached responses, and stronger projected IHVP backends that reduce per-cluster solve cost. These directions all aim to increase the feasible cache budget $C$ without the overheads of full parameter-space storage and solve costs.

CiF is *orthogonal* to prior influence accelerations (Wang et al., 2025; Schioppa et al., 2022; Koh & Liang, 2017; Klochkov & Liu, 2024). It can wrap existing IHVP solvers and curvature proxies, shifting the objective from minimizing per-query latency to minimizing *total workload cost*. The resulting bottlenecks are explicit. Cache budget $C$ is presented as a real hyperparameter. If $C$ is too small, it underfits gradient heterogeneity; if too large it erodes amortization by increasing the number of solves. Cache storage is the main large-model blocker, storing $C$ vectors in $\mathbb{R}^p$ motivates reduced precision, quantization, and structured caches that preserve linear recombination.

More broadly, CiF points toward *workload-aware* influence estimation. Allocating compute and memory over a long horizon of queries, with adaptive/hierarchical caches that refine where queries concentrate and cache representations that offer a pathway to scaling to foundation models.

## 7. Conclusion

We have introduced Clustered Influence Functions (CiF), an amortized framework that resolves the computational bottleneck of subset influence estimation by decoupling the cost of curvature inversion from the number of queries. By exploiting the low-rank structure of gradient space through sketching and clustering, CiF enables near-instantaneous approximation of arbitrary subset queries via linear recombination of cached responses. We established a rigorous error decomposition that explicitly trades off geometric fidelity against computational budget, and empirically demonstrated that CiF achieves significant speedups in high-query regimes-such as exhaustive cross-validation and large-scale data debugging-while preserving the ranking accuracy of exact solvers. This approach establishes a scalable path for integrating principled influence-based diagnostics into practical deep learning workflows.

## Acknowledgements

Support by the European Union project RRF-2.3.1-21-2022-00004 within the framework of the Artificial Intelligence National Laboratory. We acknowledge the Digital Government Development and Project Management Ltd. for awarding us access to the Komondor HPC facility based in Hungary. The authors thank Robert Bosch, Ltd. Budapest, Hungary for their generous support to the Department of Artificial Intelligence.

## Impact statement

This paper presents work whose goal is to advance the field of Machine Learning. There are many potential societal consequences of our work, none which we feel must be specifically highlighted here.

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

# Appendices

## A. Setup, notation, and basic tools

In this appendix we collect the notation and technical tools used in the analysis of clustered influence estimates. We first define the basic linearised influence-function quantities, then establish properties of the damped Gauss–Newton curvature, introduce the clustered estimator and solver residuals, describe the sampling model for folds, and finally state the Johnson–Lindenstrauss (JL) and geometric covering facts used in the error bounds.

### A.1. Data, losses, gradients, and IHVPs

We consider a supervised learning problem with training examples $(x_i, y_i) \in \mathcal{X} \times \mathcal{Y}$, $i = 1, \ldots, N$, and a parametric model $f_\theta : \mathcal{X} \to \mathbb{R}^q$ with parameters $\theta \in \mathbb{R}^p$. The per-example loss is $\ell(f_\theta(x_i), y_i)$, and the empirical training loss is

$$L(\theta) := \frac{1}{N} \sum_{i=1}^{N} \ell(f_\theta(x_i), y_i).$$

We fix a reference parameter $\theta_0$, typically an empirical risk minimiser or a late-stage iterate of a training algorithm.

For each training point we write the parameter-space gradient

$$g_i := \nabla_\theta \ell\big(f_\theta(x_i), y_i\big)\big|_{\theta=\theta_0} \in \mathbb{R}^p, \qquad i = 1, \ldots, N.$$

Let $L_{\mathrm{val}}(\theta)$ be a validation loss, and denote its gradient at $\theta_0$ by

$$g_v := \nabla_\theta L_{\mathrm{val}}(\theta)\big|_{\theta=\theta_0} \in \mathbb{R}^p.$$

To approximate the effect of removing or downweighting a subset $S \subseteq \{1, \ldots, N\}$ of size $|S| = m$, we linearise the parameter change using a damped generalised Gauss–Newton matrix $\tilde{H}_{\mathrm{GN}}$ (defined in Section A.2 below). The *exact* linearised parameter response associated with the subset $S$ is

$$\Delta\theta_S := \frac{1}{N} \tilde{H}_{\mathrm{GN}}^{-1} \sum_{i \in S} g_i \in \mathbb{R}^p. \tag{17}$$

The corresponding linearised change in validation loss is

$$\Delta\ell_{\mathrm{IF}} := g_v^\top \Delta\theta_S. \tag{18}$$

These quantities coincide with a single damped Newton/Gauss–Newton step on the linearised objective obtained by up-weighting/downweighting the subset $S$, and are standard in influence-function analyses.

Throughout, $\|\cdot\|$ denotes the Euclidean norm on vectors and the operator norm induced by it on matrices.

### A.2. Gauss–Newton curvature and damping

We recall the generalised Gauss–Newton (GN) (Schraudolph, 2002; Martens, 2020) curvature associated with the loss $L(\theta)$ and basic properties of its damped variant.

**Lemma A.1.** *Let* $L(\theta) = \frac{1}{N} \sum_{i=1}^{N} \ell(\hat{y}_i, y_i)$ *with* $\hat{y}_i = f_\theta(x_i) \in \mathbb{R}^q$. *Define the Jacobians* $J_i := \partial f_\theta(x_i)/\partial\theta \in \mathbb{R}^{q \times p}$ *and curvature matrices* $W_i := \nabla_{\hat{y}}^2 \ell(\hat{y}, y_i)\big|_{\hat{y}=f_{\theta_0}(x_i)} \succeq 0$. *Then the generalised Gauss–Newton matrix at* $\theta_0$ *is*

$$H_{\mathrm{GN}}(\theta_0) := \frac{1}{N} \sum_{i=1}^{N} J_i^\top W_i J_i \succeq 0. \tag{19}$$

*For any* $\lambda > 0$*, the* damped GN matrix

$$\tilde{H}_{\mathrm{GN}} := H_{\mathrm{GN}}(\theta_0) + \lambda I_p \tag{20}$$

*is symmetric positive definite.*

*Proof.* Linearising the model around $\theta_0$ gives $f_{\theta_0+\Delta}(x_i) \approx f_{\theta_0}(x_i) + J_i\Delta$. A second-order Taylor expansion of the loss in its first argument yields $\ell(\hat{y}_i + u, y_i) \approx \ell(\hat{y}_i, y_i) + g_i^\top u + \frac{1}{2}u^\top W_i u$, where $g_i = \nabla_{\hat{y}}\ell(\hat{y}, y_i)$ at $\hat{y} = f_{\theta_0}(x_i)$. Substituting $u = J_i\Delta$ and summing over $i$ shows that the quadratic term in $\Delta$ is $\frac{1}{2}\Delta^\top H_{\text{GN}}(\theta_0)\Delta$ with $H_{\text{GN}}(\theta_0)$ as in equation 19. Each matrix $J_i^\top W_i J_i$ is positive semidefinite because $W_i \succeq 0$, so their average is also positive semidefinite: $H_{\text{GN}}(\theta_0) \succeq 0$. Adding $\lambda I_p$ with $\lambda > 0$ yields a symmetric positive definite matrix $\tilde{H}_{\text{GN}}$. $\qquad\square$

**Lemma A.2.** *For every* $\lambda > 0$,

$$\tilde{H}_{\text{GN}} \succeq \lambda I_p, \qquad \lambda_{\min}(\tilde{H}_{\text{GN}}) \geq \lambda, \qquad \|\tilde{H}_{\text{GN}}^{-1}\| \leq \frac{1}{\lambda}. \tag{21}$$

*Proof.* By Lemma A.1, $H_{\text{GN}}(\theta_0) \succeq 0$, so for any $x \in \mathbb{R}^p$,

$$x^\top \tilde{H}_{\text{GN}} x = x^\top H_{\text{GN}}(\theta_0)x + \lambda\|x\|^2 \geq \lambda\|x\|^2.$$

This is equivalent to the matrix inequality $\tilde{H}_{\text{GN}} \succeq \lambda I_p$, and hence $\lambda_{\min}(\tilde{H}_{\text{GN}}) \geq \lambda$. For a symmetric positive definite matrix, the operator norm of the inverse satisfies $\|\tilde{H}_{\text{GN}}^{-1}\| = 1/\lambda_{\min}(\tilde{H}_{\text{GN}})$, giving $\|\tilde{H}_{\text{GN}}^{-1}\| \leq 1/\lambda$. $\qquad\square$

**Proposition A.3.** *Consider one training example* $(x, y)$, *model output* $\eta = f_\theta(x)$, *Jacobian* $J = \partial f_\theta(x)/\partial\theta$, *and loss* $\ell(\eta, y)$. *Let*

$$a := \nabla_\eta\ell(\eta, y), \qquad W := \nabla_\eta^2\ell(\eta, y).$$

*For losses arising from a correctly specified conditional likelihood with canonical parametrization, evaluated at the model distribution, we have*

$$\mathbb{E}_{y|x,\theta}[a] = 0, \qquad \mathbb{E}_{y|x,\theta}[aa^\top] = W.$$

*Consequently,*

$$\mathbb{E}_{y|x,\theta}\big[\nabla_\theta\ell(f_\theta(x), y)\nabla_\theta\ell(f_\theta(x), y)^\top\big] = J^\top W J,$$

*which is the single-example generalised Gauss–Newton summand.*

*Proof.* By the chain rule,

$$\nabla_\theta\ell(f_\theta(x), y) = J^\top a,$$

where $a = \nabla_\eta\ell(\eta, y)$ and $\eta = f_\theta(x)$. For a correctly specified conditional likelihood with canonical parametrization, the score has mean zero and its second moment equals the curvature in output space:

$$\mathbb{E}[a] = 0, \qquad \mathbb{E}[aa^\top] = W.$$

Therefore

$$\mathbb{E}_{y|x,\theta}\big[\nabla_\theta\ell(f_\theta(x), y)\nabla_\theta\ell(f_\theta(x), y)^\top\big] = J^\top\mathbb{E}[aa^\top]J = J^\top W J.$$

This is exactly the generalised Gauss–Newton summand appearing in equation 19. $\qquad\square$

## A.3. Clustered estimator and solver residuals

We now introduce the clustered approximation and the associated solver residuals. Let $\mathcal{C}$ be a partition of $\{1, \ldots, N\}$ into clusters $c$, and denote the size of cluster $c$ by $n_c$. The cluster mean gradient is

$$\mu_c := \frac{1}{n_c}\sum_{i\in c} g_i \in \mathbb{R}^p. \tag{22}$$

For each training example $i$, write its cluster index as $c(i)$ and define the deviation from the cluster mean

$$\delta_i := g_i - \mu_{c(i)}. \tag{23}$$

By construction, the deviations have zero mean within each cluster:

$$\sum_{i\in c}\delta_i = 0 \quad \text{for all } c \in \mathcal{C},$$

and hence also $\sum_{i=1}^{N} \delta_i = 0$.

The within-cluster covariance matrices are

$$\Sigma_c := \frac{1}{n_c} \sum_{i \in c} \delta_i \delta_i^\top, \qquad c \in \mathcal{C}, \tag{24}$$

and the global within-cluster covariance is

$$\bar{\Sigma} := \frac{1}{N} \sum_{i=1}^{N} \delta_i \delta_i^\top = \sum_{c \in \mathcal{C}} \frac{n_c}{N} \Sigma_c. \tag{25}$$

The scalar $\mathrm{tr}(\bar{\Sigma})$ measures the average within-cluster squared deviation.

For each cluster $c \in \mathcal{C}$ we would ideally like to compute the exact Gauss–Newton response

$$v_c^\star := \tilde{H}_{\mathrm{GN}}^{-1} \mu_c. \tag{26}$$

In practice we obtain an approximate solution $v_c$ to the linear system $\tilde{H}_{\mathrm{GN}} x = \mu_c$, for example by running a conjugate gradient solver for a finite number of steps. The residual is

$$r_c := \mu_c - \tilde{H}_{\mathrm{GN}} v_c, \qquad \rho_c := \frac{\|r_c\|}{\|\mu_c\|} \tag{27}$$

(the convention $\rho_c := 0$ when $\mu_c = 0$).

Let $S \subset \{1, \ldots, N\}$ be a subset of size $|S| = m$, and let

$$m_c(S) := \big|\{i \in S : c(i) = c\}\big|$$

be the number of selected points in cluster $c$. The *clustered, inexact* estimator of the linearised parameter response is

$$\hat{\Delta}\theta_S := \frac{1}{N} \sum_{c \in \mathcal{C}} m_c(S)\, v_c. \tag{28}$$

The corresponding approximate validation loss change is

$$\hat{\Delta}\ell := g_v^\top \hat{\Delta}\theta_S. \tag{29}$$

We will repeatedly use the following basic perturbation bound for linear systems.

**Lemma A.4.** *Let $A \in \mathbb{R}^{p \times p}$ be symmetric positive definite (SPD) and consider the system $Ax = b$. If $x_\star = A^{-1}b$ is the exact solution and $v$ is an approximate solution with residual $r := b - Av$, then*

$$x_\star - v = A^{-1}r, \qquad \|x_\star - v\| \leq \|A^{-1}\| \|r\|. \tag{30}$$

*If $A \succeq \lambda I_p$ for some $\lambda > 0$, then $\|A^{-1}\| \leq 1/\lambda$.*

*Proof.* The identity $A(x_\star - v) = b - Av = r$ implies $x_\star - v = A^{-1}r$, and taking norms yields equation 30. If $A \succeq \lambda I_p$, then $\lambda_{\min}(A) \geq \lambda$, and hence $\|A^{-1}\| = 1/\lambda_{\min}(A) \leq 1/\lambda$. $\qquad\square$

### A.4. Sampling without replacement for the fold

We assume that the subset $S \subset \{1, \ldots, N\}$ of size $|S| = m$ is drawn uniformly at random from all such subsets, independently of the clustering $\mathcal{C}$. This is the usual "simple random sample without replacement" model.

**Lemma A.5.** *Let $S$ be drawn uniformly among all subsets of $\{1, \ldots, N\}$ with $|S| = m$. For each cluster $c \in \mathcal{C}$ of size $n_c$, the count*

$$m_c(S) := \big|\{i \in S : c(i) = c\}\big| \tag{31}$$

*has a hypergeometric distribution $m_c(S) \sim \mathrm{Hypergeometric}(N, n_c, m)$ with mean*

$$\mathbb{E}\big[m_c(S)\big] = m\, \frac{n_c}{N}. \tag{32}$$

*Proof.* To obtain $m_c(S) = k$, one must choose $k$ elements of $c$ and $m - k$ elements of the complement, so

$$\Pr[m_c(S) = k] = \frac{\binom{n_c}{k}\binom{N-n_c}{m-k}}{\binom{N}{m}},$$

which is the hypergeometric distribution with population size $N$, number of "successes" $n_c$, and sample size $m$. Its mean is $m(n_c/N)$. □

The deviations $\delta_i$ defined in equation 23 have $\sum_{i=1}^{N} \delta_i = 0$. The deviation of the sampled sum from the cluster means is

$$\varepsilon_S := \sum_{i \in S} \delta_i.$$

**Lemma A.6.** *With $S$ drawn uniformly among all size-$m$ subsets of $\{1, \ldots, N\}$ and deviations $\delta_i$ satisfying $\sum_{i=1}^{N} \delta_i = 0$, the covariance of the sample mean and sample sum are*

$$\mathrm{Cov}\left(\frac{1}{m}\sum_{i \in S} \delta_i\right) = \frac{1}{m}\frac{N-m}{N-1}\bar{\Sigma}, \tag{33}$$

$$\mathrm{Cov}\left(\sum_{i \in S} \delta_i\right) = \frac{m(N-m)}{N-1}\bar{\Sigma}. \tag{34}$$

*Consequently,*

$$\mathbb{E}\|\varepsilon_S\|^2 = \frac{m(N-m)}{N-1}\,\mathrm{tr}(\bar{\Sigma}). \tag{35}$$

*Proof.* This is the standard finite-population correction for simple random sampling without replacement. The covariance of the sample mean of zero-mean population vectors $\delta_i$ is $\frac{1}{m}\frac{N-m}{N-1}\bar{\Sigma}$, see e.g. standard sampling theory texts. Multiplying by $m^2$ gives the covariance of the sample sum $\sum_{i \in S} \delta_i$. Taking the trace and using $\mathbb{E}[\varepsilon_S] = 0$ yields equation 35. □

### A.5. Base parameter and loss error bounds

We now bound the error between the exact linearised response $\Delta\theta_S$ and its clustered approximation $\hat{\Delta}\theta_S$, first deterministically and then in expectation over the sampling of $S$. These results involve only the quantities defined so far and do not require any geometric assumptions (such as JL embeddings) or probabilistic models for the gradients.

**Theorem A.7.** *Assume $\tilde{H}_{\mathrm{GN}} \succeq \lambda I_p$ with $\lambda > 0$. For any subset $S \subset \{1, \ldots, N\}$ of size $m$,*

$$\left\|\Delta\theta_S - \hat{\Delta}\theta_S\right\| \leq \frac{1}{\lambda N}\|\varepsilon_S\| + \frac{1}{\lambda N}\sum_{c \in \mathcal{C}} m_c(S)\,\rho_c\,\|\mu_c\|. \tag{36}$$

*Proof.* Decompose the sum of gradients over $S$ as

$$\sum_{i \in S} g_i = \sum_{i \in S}\left(\mu_{c(i)} + \delta_i\right) = \sum_{c \in \mathcal{C}} m_c(S)\,\mu_c + \varepsilon_S.$$

Using the definition equation 17,

$$\Delta\theta_S = \frac{1}{N}\tilde{H}_{\mathrm{GN}}^{-1}\left(\sum_{c \in \mathcal{C}} m_c(S)\,\mu_c + \varepsilon_S\right).$$

Subtracting $\hat{\Delta}\theta_S$ from equation 28 gives

$$\Delta\theta_S - \hat{\Delta}\theta_S = \frac{1}{N}\tilde{H}_{\mathrm{GN}}^{-1}\varepsilon_S + \frac{1}{N}\sum_{c \in \mathcal{C}} m_c(S)\left(\tilde{H}_{\mathrm{GN}}^{-1}\mu_c - v_c\right).$$

Taking norms and using the triangle inequality,

$$\left\|\Delta\theta_S - \hat{\Delta}\theta_S\right\| \leq \frac{1}{N}\|\tilde{H}_{\mathrm{GN}}^{-1}\|\,\|\varepsilon_S\| + \frac{1}{N}\sum_{c\in\mathcal{C}} m_c(S)\,\|\tilde{H}_{\mathrm{GN}}^{-1}\mu_c - v_c\|.$$

For each cluster $c$, Lemma A.4 applied to $A = \tilde{H}_{\mathrm{GN}}$, $b = \mu_c$, $x_\star = v_c^\star$ and $v = v_c$ yields $\|\tilde{H}_{\mathrm{GN}}^{-1}\mu_c - v_c\| = \|v_c^\star - v_c\| \leq \|\tilde{H}_{\mathrm{GN}}^{-1}\|\,\|r_c\|$, and by equation 27, $\|r_c\| \leq \rho_c\|\mu_c\|$. Using the bound $\|\tilde{H}_{\mathrm{GN}}^{-1}\| \leq 1/\lambda$ from Lemma A.2 gives equation 36. $\qquad\square$

**Theorem A.8.** *Assume $\tilde{H}_{\mathrm{GN}} \succeq \lambda I_p$ with $\lambda > 0$, and let $S$ be drawn uniformly among all subsets of $\{1, \ldots, N\}$ of size $m$, independently of the clustering $\mathcal{C}$. Then*

$$\mathbb{E}\left\|\Delta\theta_S - \hat{\Delta}\theta_S\right\| \leq \frac{1}{\lambda N}\sqrt{\frac{m(N-m)}{N-1}\,\mathrm{tr}(\bar{\Sigma})} + \frac{m}{\lambda N}\sum_{c\in\mathcal{C}}\frac{n_c}{N}\,\rho_c\,\|\mu_c\|. \tag{37}$$

*The expectation is taken over the random choice of $S$.*

*Proof.* Taking expectations in equation 36 and applying Jensen's inequality to the first term yields

$$\mathbb{E}\left\|\Delta\theta_S - \hat{\Delta}\theta_S\right\| \leq \frac{1}{\lambda N}\mathbb{E}\|\varepsilon_S\| + \frac{1}{\lambda N}\sum_{c\in\mathcal{C}}\mathbb{E}\big[m_c(S)\big]\rho_c\,\|\mu_c\|.$$

By Jensen, $\mathbb{E}\|\varepsilon_S\| \leq \sqrt{\mathbb{E}\|\varepsilon_S\|^2}$, and Lemma A.6 gives $\mathbb{E}\|\varepsilon_S\|^2 = \frac{m(N-m)}{N-1}\,\mathrm{tr}(\bar{\Sigma})$. For the second term, Lemma A.5 ensures $\mathbb{E}[m_c(S)] = m(n_c/N)$. Substituting these expressions into the previous inequality yields equation 37. $\qquad\square$

**Theorem A.9.** *Under the assumptions of Theorem A.8, the expected difference between the exact and clustered linearised validation loss changes satisfies*

$$\boxed{\begin{aligned} \mathbb{E}\big|\Delta\ell_{\mathrm{IF}} - \hat{\Delta}\ell\big| &\leq \frac{\|g_v\|}{\lambda N}\sqrt{\frac{m(N-m)}{N-1}\,\mathrm{tr}(\bar{\Sigma})} \\ &\quad + \frac{\|g_v\|}{\lambda N}m\sum_{c\in\mathcal{C}}\frac{n_c}{N}\,\rho_c\,\|\mu_c\|. \end{aligned}} \tag{38}$$

*Proof.* By definition, $\Delta\ell_{\mathrm{IF}} - \hat{\Delta}\ell = g_v^\top(\Delta\theta_S - \hat{\Delta}\theta_S)$, so by Cauchy–Schwarz,

$$\big|\Delta\ell_{\mathrm{IF}} - \hat{\Delta}\ell\big| \leq \|g_v\|\,\big\|\Delta\theta_S - \hat{\Delta}\theta_S\big\|.$$

Taking expectations and using Theorem A.8 yields equation 38. $\qquad\square$

Theorem A.9 is the basic inequality underlying the later error bounds: it separates the expected loss error into a clustering term, controlled by $\mathrm{tr}(\bar{\Sigma})$, and a solver term, controlled by the residuals $\rho_c$ and norms $\|\mu_c\|$.

## A.6. Johnson–Lindenstrauss maps and projected covariance

We next introduce a Johnson–Lindenstrauss (JL) embedding (Johnson et al., 1984), which allows us to relate the within-cluster covariance $\bar{\Sigma}$ to a lower-dimensional projected analogue.

**Definition A.10.** Let $\Pi \in \mathbb{R}^{d\times p}$ be a linear map. We say that $\Pi$ is a JL embedding with distortion parameter $\varepsilon_{\mathrm{JL}} \in (0,1)$ for a finite set of vectors $\mathcal{U} \subset \mathbb{R}^p$ if with a very high probability

$$(1 - \varepsilon_{\mathrm{JL}})\,\|u\|^2 \leq \|\Pi u\|^2 \leq (1 + \varepsilon_{\mathrm{JL}})\,\|u\|^2 \qquad \text{for all } u \in \mathcal{U}. \tag{39}$$

In our setting we will apply such a map to the deviations $\delta_i$ in equation 23. Define the projected deviations $\Pi\delta_i \in \mathbb{R}^d$, and the corresponding covariance

$$\bar{\Sigma}^\Pi := \frac{1}{N}\sum_{i=1}^{N}(\Pi\delta_i)(\Pi\delta_i)^\top. \tag{40}$$

**Lemma A.11.** *Suppose* $\Pi \in \mathbb{R}^{d \times p}$ *is a JL embedding with distortion parameter* $\varepsilon_{\mathrm{JL}} \in (0,1)$ *for the set* $\{\delta_1, \ldots, \delta_N\}$. *Then*

$$(1 - \varepsilon_{\mathrm{JL}}) \operatorname{tr}(\bar{\Sigma}) \; \leq \; \operatorname{tr}(\bar{\Sigma}^\Pi) \; \leq \; (1 + \varepsilon_{\mathrm{JL}}) \operatorname{tr}(\bar{\Sigma}). \tag{41}$$

*Proof.* By definition, $\operatorname{tr}(\bar{\Sigma}) = \frac{1}{N} \sum_{i=1}^{N} \|\delta_i\|^2$ and $\operatorname{tr}(\bar{\Sigma}^\Pi) = \frac{1}{N} \sum_{i=1}^{N} \|\Pi \delta_i\|^2$. The inequalities equation 41 follow immediately by applying equation 39 with $u = \delta_i$ and averaging over $i$. $\qquad\square$

# B. Extended results from main-text

### B.1. Robustness: ablations over $(C, d_\Pi)$ on MNIST 10% (LeNet)

**Ablation summary.** Figure 5 shows the Pearson and Spearman correlation between IF and CiF across varying numbers of folds $K$, JL dimensions, and cluster counts. For moderate to large fold counts ($K = 100, 250, 500$), CiF exhibits consistently high agreement with IF: rank correlations typically exceed $0.92$ and often approach $0.98$–$0.99$ across all tested JL dimensions ($32$–$256$) and cluster counts ($10$–$500$). This indicates that once the IF target is stabilized by sufficient fold averaging, neither aggressive JL compression nor substantial clustering degrades ranking fidelity. In contrast, the $K = 5$ regime shows visibly lower and more variable correlations—particularly in Spearman—suggesting that small-$K$ settings provide a weak and noisy reference and are therefore unsuitable for assessing robustness.

**Implications.** Overall, the ablations reveal rapid saturation of ranking agreement: a JL dimension as low as 32 and moderate clustering already achieve near-optimal performance, while increasing JL dimension or the number of clusters primarily affects scale calibration rather than rank ordering. Increasing the number of folds beyond $K \approx 100$ yields diminishing returns, with only marginal changes in correlation. These results support the conclusion that CiF reliably preserves IF rankings in realistic cross-validation settings, and that strong agreement emerges well before computationally expensive or high-dimensional configurations are required.

Figures 6 and 7 show the full correlation matrices for all selected settings on MNIST 10% (LeNet) with $K = 50$ and $K = 500$ folds respectively.

# C. Streamed Gaussian JL projection

For large networks, the parameter dimension $p$ makes it infeasible to store the full per-example gradient matrix $G \in \mathbb{R}^{N \times p}$, and it is equally impractical to materialize an explicit Gaussian JL matrix $P \in \mathbb{R}^{d \times p}$ when $p$ is large. Nevertheless, clustering requires a per-example feature representation derived from gradients. We therefore compute the JL features *in a streaming manner*: we visit coordinates of each gradient vector sequentially and accumulate the projected features as a streamed linear combination.

**Streaming as a linear sketch.** Fix a Gaussian projection

$$P \in \mathbb{R}^{d \times p}, \qquad P_{ab} \sim \mathcal{N}\left(0, \frac{1}{d}\right),$$

and define the linear map $\varphi : \mathbb{R}^p \to \mathbb{R}^d$ by $\varphi(x) = Px$. For any vector $x \in \mathbb{R}^p$ we wish to compute $y = \varphi(x)$ without storing $P$ and without holding all of $x$ in memory at once. Partition the coordinate set $\{1, \ldots, p\}$ into contiguous blocks $\mathcal{I}_1, \ldots, \mathcal{I}_B$ (e.g., $\mathcal{I}_b = \{s_b, \ldots, e_b - 1\}$). Write $x = \sum_{b=1}^{B} x_{\mathcal{I}_b}$ where $x_{\mathcal{I}_b} \in \mathbb{R}^p$ is supported only on coordinates in $\mathcal{I}_b$. By linearity,

$$y = Px = \sum_{b=1}^{B} Px_{\mathcal{I}_b} = \sum_{b=1}^{B} P_{:,\mathcal{I}_b} x_{\mathcal{I}_b},$$

where $P_{:,\mathcal{I}_b} \in \mathbb{R}^{d \times |\mathcal{I}_b|}$ denotes the restriction of $P$ to columns in $\mathcal{I}_b$ and $x_{\mathcal{I}_b} \in \mathbb{R}^{|\mathcal{I}_b|}$ is the corresponding subvector. This yields a streamed accumulation rule:

$$y^{(0)} = 0, \qquad y^{(b)} = y^{(b-1)} + P_{:,\mathcal{I}_b} x_{\mathcal{I}_b} \quad (b = 1, \ldots, B),$$

and $y^{(B)} = Px$. At any time, the working state is just the accumulator $y^{(b)} \in \mathbb{R}^d$ plus the current block $x_{\mathcal{I}_b}$ and the transient random block $P_{:,\mathcal{I}_b}$.

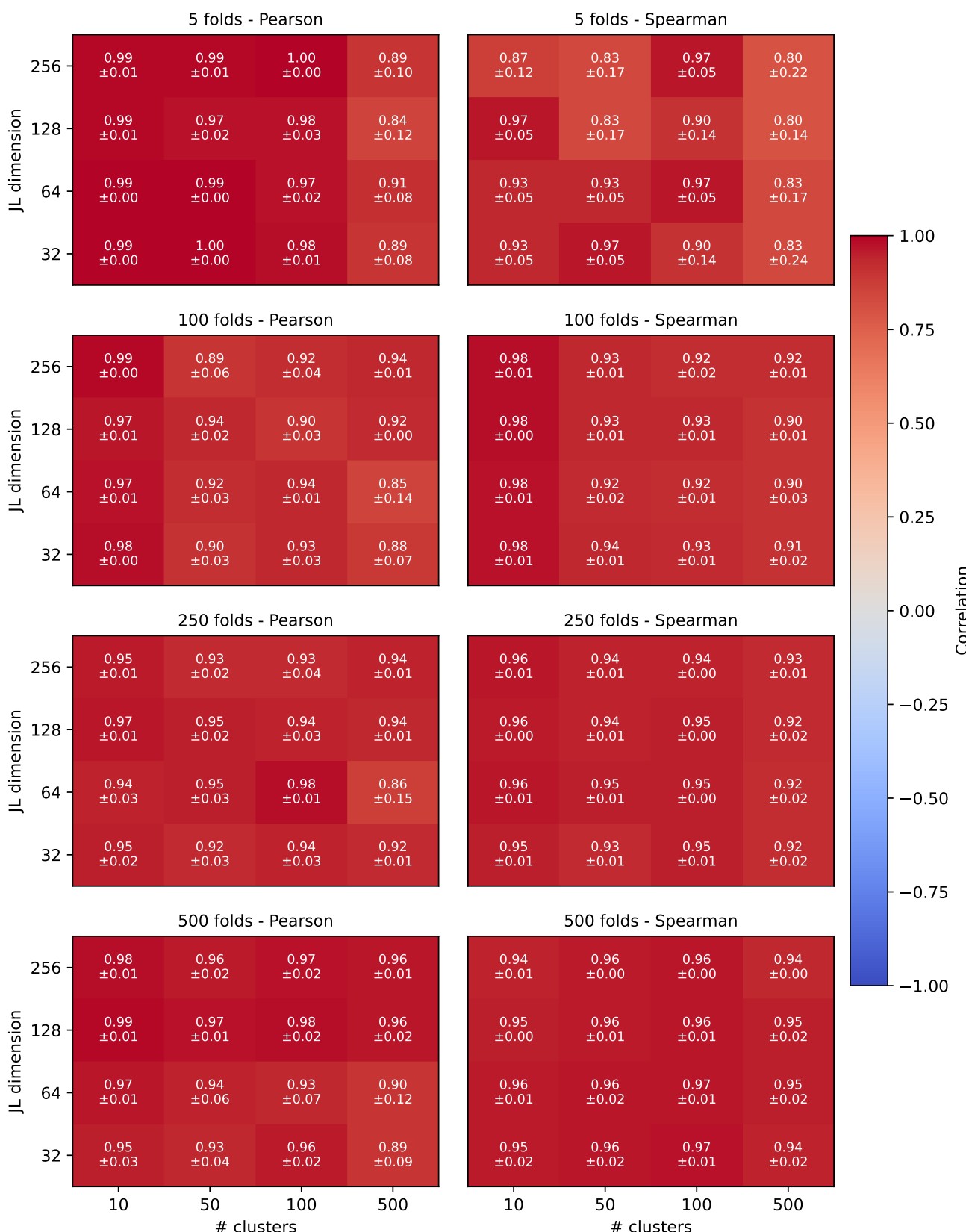

*Figure 5.* **Robustness ablation on MNIST 10% (LeNet): CiF vs IF.** Heatmaps show correlation between CiF and the per-fold IF baseline across fold queries, as a function of cluster budget $C$ (x-axis) and JL sketch dimension $d_\Pi$ (y-axis). Folds: $\in \{5, 100, 250, 500\}$, Left: Pearson. Right: Spearman.

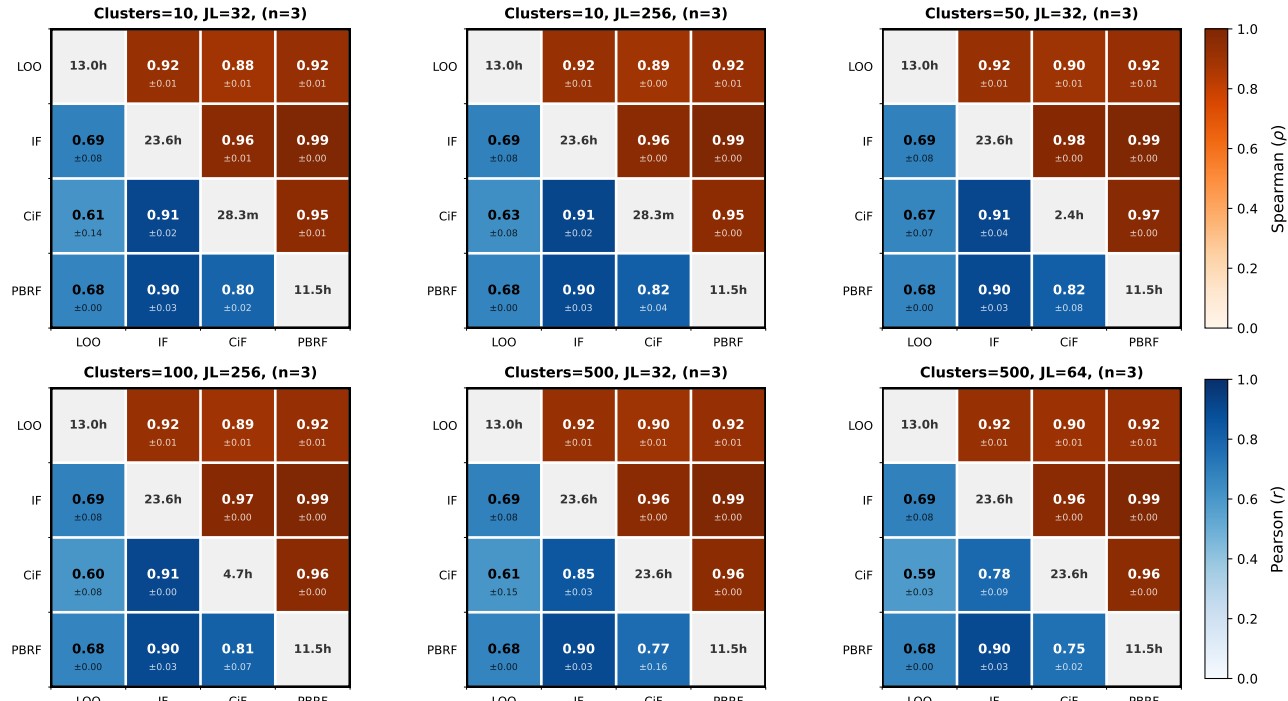

*Figure 6.* **Correlation matrices for fold-level $\Delta$loss predictions (MNIST-10% LeNet; $K=500$).** Rows/columns compare leave-one-out retraining (LOO), influence functions (IF), our cached clustered estimator (CiF), and the proximal Bregman response function (PBRF). Each off-diagonal cell shows *upper*: Spearman $\rho$ and *lower*: Pearson $\rho$ over the $K$ folds; color encodes correlation. Diagonal entries report end-to-end runtimes $T$.

**Per-sample gradient features for clustering.** Let $\theta \in \mathbb{R}^p$ be the model parameters and $\ell_i(\theta)$ the per-example loss. Define the per-example gradient

$$g_i \triangleq \nabla_\theta \ell_i(\theta) \in \mathbb{R}^p, \qquad i \in \{1, \ldots, N\}.$$

We construct the clustering features by applying the same linear sketch $\varphi$ to each gradient:

$$z_i \triangleq \varphi(g_i) = Pg_i \in \mathbb{R}^d.$$

Crucially, a *single* projection matrix $P$ is shared across all examples, so the embedded vectors $\{z_i\}$ lie in a common coordinate system and pairwise distances satisfy

$$\|z_i - z_j\|_2 = \|P(g_i - g_j)\|_2,$$

which is the standard JL embedding of gradient differences used by the clustering objective.

**Dataset-level streaming.** Streaming is performed at two levels: (i) within each vector $g_i$, we compute $z_i = Pg_i$ by the blockwise accumulation above, and (ii) across the dataset, we process samples one-by-one to form the matrix

$$Z \triangleq \begin{bmatrix} z_1^\top \\ \vdots \\ z_N^\top \end{bmatrix} \in \mathbb{R}^{N \times d} \qquad \text{without ever materializing } G \in \mathbb{R}^{N \times p}.$$

The result is an $N \times d$ JL feature matrix suitable for clustering while keeping memory bounded independently of $p$ (up to the chosen block size and the $d$-dimensional accumulator).

**Implicitly fixed projection.** Although $P$ is not stored explicitly, it is conceptually fixed. Equivalently, one may view $P$ as generated deterministically from a seed, so that each column $P_{:t}$ is a fixed pseudorandom function of the seed and index $t$. Streaming simply evaluates the needed columns in blocks and accumulates the linear combination, producing exactly the same $z_i = Pg_i$ as an explicit dense multiplication would.

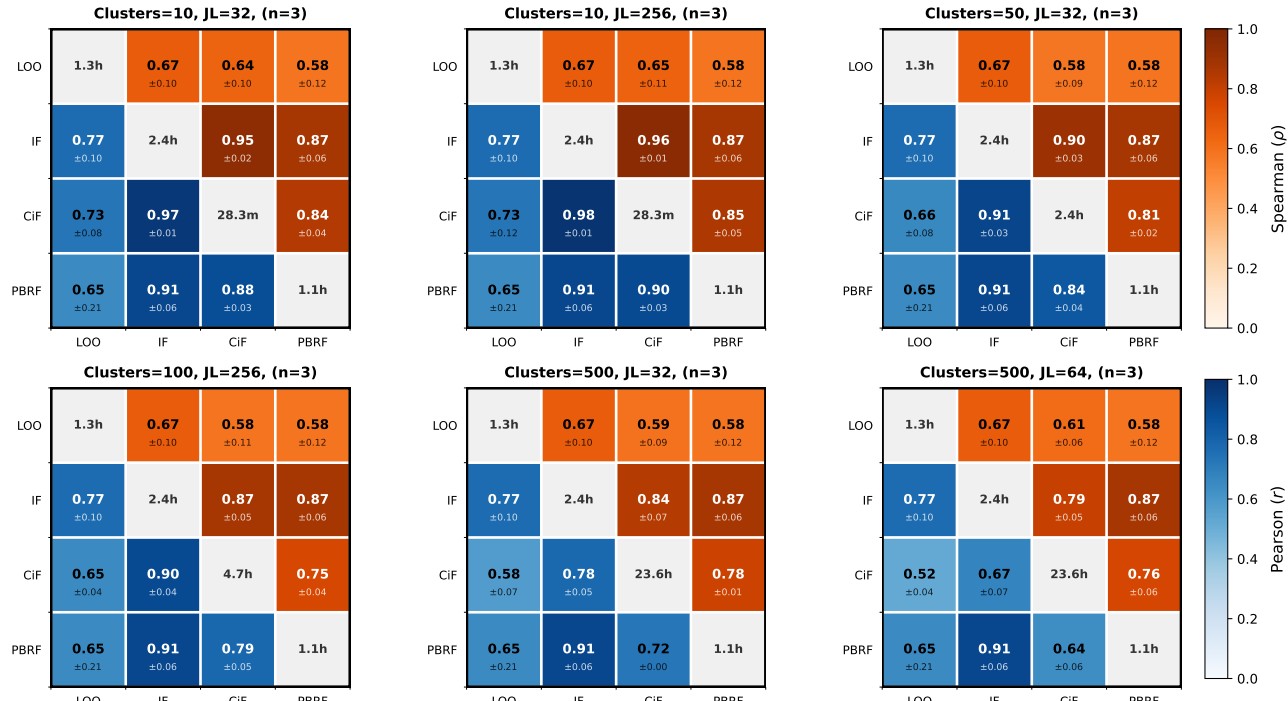

*Figure 7.* **Correlation matrices for fold-level Δloss predictions (MNIST-10% LeNet; $K$=50).** Rows/columns compare leave-one-out retraining (LOO), influence functions (IF), our cached clustered estimator (CiF), and the proximal Bregman response function (PBRF). Each off-diagonal cell shows *upper*: Spearman $\rho$ and *lower*: Pearson $\rho$ over the $K$ folds; color encodes correlation. Diagonal entries report end-to-end runtimes $T$.

## D. End-to-end runtime and memory footprint

Table 4 profiles resource trade-offs across architectures. Cache Storage denotes the persistent footprint for influence vectors ($4Cp$ bytes), excluding transient solver memory ($\approx 6p$) incurred by both methods. The Offline Build breakdown reveals that geometric overhead (Sketching + Clustering) remains constant per dataset (e.g., $\approx 1.2$h for AlexNet), whereas solve costs scale linearly. Empirically, MNIST-10% results verify the amortization frontier: at $C = 500$, build time converges to baseline runtime (23.6h), confirming net acceleration whenever query volume $Q > C$.

*Table 4.* **System Profile.** We compare the persistent **Cache Storage** requirement (scaling as $O(Cp)$) against runtime performance. Standard IF requires zero storage but incurs high online costs. CiF trades moderate storage (e.g., 12.2 GB for AlexNet) for near-instantaneous **Online Latency** ($< 0.1$s). The **Offline Overhead** (Projection + Clustering) remains negligible and constant across cluster sizes, confirming that the setup cost is decoupled from the cluster budget.

| Exp. Setting | Method | Cache ($C$) | Cache Storage | Offline Build Phase | | | Online Phase | |
|---|---|---|---|---|---|---|---|---|
| | | | | Overhead | IHVP Solves | Total Build | Per Query | Total Time |
| **ImageNet-1k** | IF | – | 0 | – | – | – | $\approx 7.5$ min | $\approx 625.0$ h |
| DeiT-Tiny ($p \approx 5.7$M) | CiF (Ours) | 512 | 11.7 GB | 3.0 h | 64.0 h | 67.0 h | $< 0.1$ s | $< 0.1$ h |
| ($K$=5000) | CiF (Ours) | 128 | 2.9 GB | 3.0 h | 16.0 h | 19.0 h | $< 0.1$ s | $< 0.1$ h |
| **CIFAR-10** | IF | – | 0 | – | – | – | 30.0 min | 75.0 h |
| ResNet-18 ($p \approx 11$M) | CiF (Ours) | 50 | 2.2 GB | 0.4 h | 25.0 h | 25.4 h | $< 0.1$ s | $< 0.1$ h |
| ($K$=150) | CiF (Ours) | 10 | 0.5 GB | 0.4 h | 5.0 h | 5.4 h | $< 0.1$ s | $< 0.1$ h |
| **CIFAR-10** | IF | – | 0 | – | – | – | 25.3 min | 63.3 h |
| AlexNet ($p \approx 61$M) | CiF (Ours) | 50 | 12.2 GB | 1.2 h | 21.1 h | 22.3 h | $< 0.1$ s | $< 0.1$ h |
| ($K$=150) | CiF (Ours) | 10 | 2.4 GB | 1.2 h | 4.2 h | 5.4 h | $< 0.1$ s | $< 0.1$ h |
| **CIFAR-10** | IF | – | 0 | – | – | – | 12.8 min | 32.1 h |
| LeNet ($p \approx 60$k) | CiF (Ours) | 50 | 13 MB | 5 min | 10.7 h | 10.7 h | $< 0.01$ s | $< 0.1$ h |
| ($K$=150) | CiF (Ours) | 10 | 2.5 MB | 5 min | 2.1 h | 2.2 h | $< 0.01$ s | $< 0.1$ h |
| | IF | – | 0 | – | – | – | 2.8 min | 23.6 h |
| **MNIST-10%** | CiF (Ours) | 10 | 2.5 MB | $< 1$ min | 28.3 min | 28.3 min | $< 0.01$ s | $< 0.1$ min |
| LeNet ($p \approx 60$k) | CiF (Ours) | 50 | 12.5 MB | $< 1$ min | 2.4 h | 2.4 h | $< 0.01$ s | $< 0.1$ min |
| ($K$=500) | CiF (Ours) | 100 | 25.0 MB | $< 1$ min | 4.7 h | 4.7 h | $< 0.01$ s | $< 0.1$ min |
| | CiF (Ours) | 500 | 125.0 MB | $< 1$ min | 23.6 h | 23.6 h | $< 0.01$ s | $< 0.1$ min |

