# OpenReview forum: "Clustered Influence Functions"
_ICML.cc/2026/Conference — ICML 2026 regular_

### Official Review · Reviewer_uw4x · 2026-03-09

**Soundness:** 3
**Presentation:** 4
**Significance:** 3
**Originality:** 3
**Overall Recommendation:** 3
**Confidence:** 2

**Summary:**

This paper proposes Clustered Influence Functions (CiF), an acceleration method for influence function (IF) computation. The key idea is to cluster per-example gradients and amortize inverse-Hessian-vector product (IHVP) computations across cluster representatives, reducing cost when evaluating many subset queries. The method is evaluated on MNIST, CIFAR-10, and ImageNet with runtime–fidelity trade-off analyses.

**Compliance With Llm Reviewing Policy:**

Affirmed.

**Key Questions For Authors:**

1. Could the authors comment on whether there are dataset- or model-dependent diagnostics that could help practitioners anticipate when CiF is likely to maintain acceptable fidelity? In particular, are there structural properties of gradient distributions (e.g., clustering tendency, intrinsic dimension) that determine when the clustering approximation is reliable?



2. The method exposes a speed–fidelity trade-off controlled by the cluster budget C.

Is there any principled way to select C before running the full IF computation, or to estimate the expected fidelity degradation given the gradient distribution of a dataset?


3. The method relies on standard clustering techniques (e.g., k-means) applied to gradient representations.

Do the authors believe the effectiveness of CiF primarily depends on properties of the gradient geometry (e.g., natural clustering structure)? If so, it would be helpful to understand under what conditions this assumption is expected to hold.

**Limitations:**

yes

**Strengths And Weaknesses:**

## Strengths

### 1. Clean and well-written

The paper is clearly structured and easy to follow.
The motivation (amortizing IF computation across repeated subset queries) aligns well with the proposed method, and the empirical evaluation directly reflects this motivation. The simulation results are tightly coupled with the methodological claims, and the overall narrative is coherent and logically organized. The presentation makes the trade-offs transparent and easy to interpret.

---

### 2. Simplicity and applicability of the method

The method is conceptually simple: clustering gradients and reusing IHVP computations at the cluster level. Because of this simplicity, it appears broadly applicable across different settings. The approach is not tied to a specific architecture, task (e.g., classification), or inference scenario. This generality and low conceptual overhead make the method easy to integrate into existing IF pipelines.

---

### 3. Clear and well-characterized trade-off

The paper clearly presents the speed–fidelity trade-off inherent in the approach.
The methodological design and the experiments are well aligned: the reader can anticipate how varying the cluster budget \(C\) affects runtime and ranking fidelity, and the empirical results reflect this behavior. The phase diagrams and correlation tables effectively illustrate where the method performs well and where it degrades. This clarity in exposing trade-offs is a strong aspect of the paper.

---

## Weaknesses

### 1. Limited conceptual and technical novelty

The method combines existing components:

- k-means clustering on gradients,
- reuse of IHVP computations across cluster means,
- amortization analysis of computational cost.

While cleanly integrated, the approach does not appear to introduce a fundamentally new conceptual or theoretical insight. The observed speedup scales approximately linearly in \(Q/C\), which is structurally predictable from the amortization design.

The paper reports Pearson and Spearman correlations as primary validation metrics. However, the novel contribution lies in applying clustering to IF computation and amortizing IHVP, while the clustering mechanism itself relies on standard methods such as k-means. The resulting correlation values therefore seem largely driven by:

- the intrinsic structure of the dataset (e.g., low-rank or clustered gradient behavior), and
- the quality of k-means clustering,

rather than by a new theoretical guarantee introduced by CiF itself.

In this sense, high Pearson/Spearman correlation in small-scale or structured datasets appears more as a property of the data regime than as a methodological breakthrough. These metrics should therefore be interpreted as regime-dependent validation rather than as evidence of strong novel fidelity guarantees.

---

### 2. Limited practical regime in large-scale settings

The primary selling point of the method is acceleration. However, in the large-scale regime where acceleration is most critical (e.g., ImageNet-1k with DeiT-Tiny), ranking fidelity degrades substantially (e.g., ρ ≈ 0.07–0.19).

This creates a structural tension:

- In small or structured datasets, fidelity is high but acceleration is less critical.
- In large, high-dimensional datasets where acceleration is most valuable, fidelity significantly deteriorates.

As currently presented, the method appears least reliable in the regime where its computational advantages matter most. This limits its practical applicability in realistic large-scale settings.

Moreover, while this behavior reflects an inherent speed–fidelity trade-off, the paper does not provide a principled mechanism to anticipate or control this trade-off in advance. The cluster budget \(C\) functions as a tuning knob, but there is no structural guidance (e.g., diagnostics, error indicators, or predictive criteria) that allows practitioners to determine when fidelity degradation will become prohibitive in a given dataset or model. As a result, the method’s reliability in large-scale heterogeneous settings remains difficult to assess a priori.

---

## Overall Assessment

The paper is technically sound, clearly written, and presents a clean and simple acceleration strategy. The trade-offs are transparently analyzed, and the method is easy to understand and potentially easy to deploy.

However, the resulting gain appears relatively modest compared to the simplicity of the approach. While the method could be useful in some practical scenarios, its applicability appears restricted in large-scale settings where such techniques would be most needed.

Overall, I view this as a solid but incremental engineering contribution whose impact may be limited in realistic large-scale scenarios.

---

## Score

**3 — Borderline Reject**

Sound and well-executed, but limited novelty and restricted practical regime in large-scale settings.

---

> ### Author Rebuttal · Authors · 2026-03-30
>
> We thank the reviewer for the thorough and fair assessment. The review correctly identifies the core speed–fidelity tradeoff and acknowledges the paper is technically sound, clearly written, and transparently presents its limitations. We address each concern below.
>
> > W1: the approach does not appear to introduce a fundamentally new conceptual or theoretical insight... high Pearson/Spearman correlation... appears more as a property of the data regime than as a methodological breakthrough:
>
> We partially agree that correlation values are regime-dependent, and this is what Theorem 3.1 predicts. The novelty of CiF is not k-means or IHVP reuse individually. It is the observation that every subset query S enters the influence equation only through the gradient sum $\sum_{i\in S}g_i$ (Eq. 3), meaning all Q queries share the same operand structure. This means expensive curvature inversion can be decoupled from the query counts entirely: pay C solves offline, answer Q queries online via linear recombination. Prior acceleration methods (FastIF, TracIn, LoGra, TRAK) reduce the cost of individual IHVP solves but still pay per query. CiF eliminates per-query solves altogether.
>
> Theorem 3.1 then formalizes exactly how much approximation error this introduces, decomposing it into a clustering scatter term controlled by tr($\Sigma$), the average within-cluster gradient variance, and a solver residual term controlled by CG tolerance $p_c$.
>
> > W2: the method appears least reliable in the regime where its computational advantages matter most:
>
> We respectfully disagree. The argument implicitly treats C as fixed, but C is a free hyperparameter with a monotone effect on fidelity. Theorem 3.1 makes this explicit: the clustering scatter term decreases as C grows, because larger C produces tighter clusters and consequently smaller tr($\Sigma$). The experiments also confirm this. With every scenario, a higher C increased the fidelity. The tradeoff is not a structural limitation of the method, rather a compute budget decision. A practitioner with a large offline budget can increase C, consequently, reduce tr($\Sigma$), and receive a higher fidelity. But if someone doesn’t have a big compute budget, and low fidelity is sufficient, CiF also works in that scenario, and Theorem 3.1 predicts the expected fidelity drop that should be anticipated.
>
> > Q1/Q3 Are there dataset- or model-dependent diagnostics that could help practitioners anticipate when CiF maintains fidelity? Are there structural properties of gradient distributions that determine when the clustering approximation is reliable?
>
> Yes, and Theorem 3.1 gives the answer directly: CiF achieves high fidelity when tr($\Sigma$) is small. Two model/dataset conditions support this. First, neural collapse: during the terminal phase of training, within-class gradient variation collapses and class-mean gradient dominate, directly bounding within-cluster scatter when clusters align with class structure. Second, low effective gradient rank: the JL projection (of low effective rank) is nearly lossless, so clustering in the projected space faithfully recovers the full-space partition. Practically, tr($\Sigma$) is computable from the cache construction itself and servers as an a priori diagnostic, a high value signals C is too small or the dataset/model pair before any queries are run.
>
> > Q2: Is there any principled way to select C?
>
> Yes. We refer to our response to reviewer svkd’s Q2.

---

> > ### Author Rebuttal · Reviewer_uw4x · 2026-04-03
> >
> > Thank you for the rebuttal. One follow-up on the tr(Σ̄) diagnostic claim.
> > The rebuttal states that tr(Σ̄) is computable from the cache construction itself and serves as an a priori diagnostic — a high value signals C is too small before any queries are run. We find this claim interesting, but would like to understand its practical scope more precisely.
> > Can tr(Σ̄) be used as a practical guideline for selecting C in a new large-scale setting — i.e., does it provide a principled way to obtain a concrete C value prior to running experiments? If so, two concrete questions follow:
> >
> > What is the recommended procedure? For instance, does one sweep C, compute tr(Σ̄) at each value, and stop when tr(Σ̄) falls below some threshold? If so, how is that threshold determined without ground-truth fidelity?
> > Applying this to the ImageNet/DeiT-Tiny setting: given the reported tr(Σ̄) values (if available) at C ∈ {128, 512}, what would the adequate C be to achieve practically meaningful rank correlation, and is this C computationally feasible given the reported build times?
> >
> > If tr(Σ̄) functions primarily as a post-hoc diagnostic rather than a selection criterion, that would be a useful clarification — but it would also weaken the claim that CiF provides principled guidance for large-scale deployment, which is central to addressing W2.

---

> > > ### Author Response · Authors · 2026-04-07
> > >
> > > Thank you for the follow-up. The intended claim is weaker than a universal threshold rule, but it is stronger than a post-hoc explanation: tr$(\Sigma)$ is a pre-solve budget diagnostic. It can be computed after Phase I (sketching + clustering) and before any IHVP solves are run, so its practical role is to compare candidate C values and detect when the currently feasible cache budget is clearly too small.
> > >
> > > Concretely, the procedure we recommend is:
> > >
> > > 1.	Choose a feasible grid of candidate C values from a memory / offline budget.
> > >
> > > 2.	Run the sketching once, and do the clustering for each candidate C.
> > >
> > > 3.	For each candidate C, compute the tr$(\Sigma)$ for each.
> > >
> > > 4.	Plot and inspect the tr$(\Sigma)$ vs C curve, and select the smallest C beyond which the marginal decrease begins to flatten
> > >
> > > So the intended use is similar to an elbow/scree diagnostic: it provides a relative ranking of cache budgets before the expensive solve phase. We do not claim a universal threshold on tr$(\Sigma)$ that guarantees a target Pearson/Spearman value without any calibration, in that sense, it is not a fidelity-calibrated selector, but it is still actionable because it can rule out under provisioned cache sizes before paying for many solves.
> > >
> > >  Applied to ImageNet/DeiT-Tiny, we do not think it would be responsible to claim a precise deployable C from the two reported points. The only conclusion supported by the current evidence is the negative one (that was presented in the paper): these budgets are still in high-scatter regime, so the tested cache sizes remain too small for high-fidelity deployment in that setting. Thus, the diagnostic is useful there mainly as a warning signal that substantially larger C would be needed. Whether that larger C is computationally attractive is the practical limitation which we highlight by the stress test.
> > >
> > > So to state the claim precisely: tr$(\Sigma)$ should be interpreted as a pre-solve screening / model-selection diagnostic over candidate C values, not as a universal closed-form rule that outputs the correct deployable C in every new large-scale regime. We will clarify this distinction more explicitly in the paper.

---

### Official Review · Reviewer_svkd · 2026-03-12

**Soundness:** 3
**Presentation:** 3
**Significance:** 3
**Originality:** 3
**Overall Recommendation:** 4
**Confidence:** 3

**Summary:**

This paper addresses the computational bottleneck of subset influence estimation under high-query workloads. It introduces Clustered Influence Functions (CiF), which amortize influence computation by building an offline cache from clustered training gradients and reusing cached curvature-based responses to answer arbitrary subset queries via linear recombination. The method combines gradient sketching, clustering, and damped GGN solves, and is accompanied by an error analysis that separates clustering approximation error from solver residual error. Experiments on MNIST, CIFAR-10, and ImageNet-scale settings show that CiF can substantially reduce runtime while maintaining strong agreement with standard influence-based rankings, especially when many subset queries must be answered. Overall, the paper proposes a practical and scalable framework for influence-based data analysis in deep learning.

**Compliance With Llm Reviewing Policy:**

Affirmed.

**Final Justification:**

The rebuttal is clear and constructive, and it adequately addresses my main concerns. The authors clarify the role of gradient clusterability in approximation quality, explain the main bottlenecks at larger scale, and provide practical guidance for choosing key hyperparameters. The discussion of cache validity under model updates is also helpful and makes the intended operating regime of the method much clearer.

Overall, my main concerns have been resolved after rebuttal, and I maintain my score.

**Key Questions For Authors:**

1. The ImageNet-scale results show strong speedups but lower agreement with standard IF. Could the authors clarify what they believe is the main bottleneck there: cache size, weaker gradient cluster structure, or solver approximation?

2. Could the authors provide a practical guideline for choosing 𝐶, sketch dimension, and solver tolerance, and for deciding when a workload is large enough for cache construction to be worthwhile?

3. How sensitive is the cache to small model updates? A brief discussion of cache reuse or invalidation under mild model drift would strengthen the practical picture.

**Limitations:**

Yes

**Strengths And Weaknesses:**

This paper is technically solid and addresses an important practical bottleneck of influence-function methods in high-query subset settings. The proposed CiF framework is well motivated, clearly formulated, and supported by a useful offline/online design, an interpretable error decomposition, and experiments across MNIST, CIFAR-10, and ImageNet-scale settings. I also think the paper is reasonably original: while the individual ingredients are not new, the main contribution is the reframing of subset influence as an amortized workload with reusable cache-based computation. The paper is also generally well written and easy to follow.

The main weaknesses are that approximation quality depends on how well gradients cluster, and performance appears less convincing in larger, more complex settings where agreement with standard IF can drop. The method also introduces nontrivial cache storage overhead, which may limit scalability for very large models. Overall, I find the paper sound, clearly presented, and practically meaningful, with the main limitations lying in approximation fidelity and storage tradeoffs at larger scale.

---

> ### Author Rebuttal · Authors · 2026-03-30
>
> We thank the reviewer for the thorough and positive assessment. The reviewer correctly identifies the main contribution, reframing subset influence as an amortized workload with reusable cache-based computation. We address the questions and weaknesses below.
>
> > W1: Approximation quality depends on the clusterability of gradients:
>
> Yes, and this is explicitly characterized by Theorem 3.1. The expected error is controlled by tr($\Sigma$), the within-cluster scatter of the gradients. When gradients cluster well, tr($\Sigma$) is small and fidelity is high. When gradients are heterogeneous tr($\Sigma$) is large and more clusters are needed.
>
> > W2/Q1: Performance less convincing at larger scale, what is the main bottleneck at ImageNet?
>
> The main bottleneck is cache size. C$\in${128,512 for 1.28M examples is roughly 10000:1 compression, tr($\Sigma$) remains large at this extreme simply because no fixed small set of clusters can faithfully represent 1.28M heterogeneous gradients. The gradient structure is also weaker at ImageNet, than on CIFAR-10 or MNIST, within-class diversity is higher, so gradients do not collapse toward class means as cleanly.
>
> > W3: Storage overhead:
>
> Acknowledged in the paper. We refer our response to reviewer hXq6 L1.
>
> > Q2: Practical guideline for choosing C, sketch dimension, solver tolerance, and workload threshold:
>
> Choosing C: This can be determined utilizing Theorem 3.1. After Phase I (sketching and clustering), tr($\Sigma$) is computable at O(Np) cost before any IHVP solve. Run k-means for a range of C values, compute tr($\Sigma$) for each, and plot scatter vs C. Pick the elbow where scatter saturates.
>
> Choosing the sketch dimension: Because the sketch only affects the clustering partition (cluster means and all curvature solves are computed in the original parameter space), we recommend a moderately small dimension $d_\Pi \in$[32,128]. This balances the compression fidelity against the clustering cost. Aggressive compression does not hurt ranking fidelity because the sketching is only used for the clustering, all curvature solves are done in full parameter space.
>
> Choosing the solver tolerance: Choose the tolerance so that the solver term is << then the clustering term, in Theorem 3.1. This connects the tolerance choice directly to the error bound and ensures the clustering approximation is the dominant source of error, not the solver.
>
> > Q3: Cache sensitivity to small model updates:
>
> The cache is tied to a fixed $\theta_0$, specifically, the cluster means $\mu_c$ are gradients at $\theta_0$, and the cached vectors $v_c=H^{-1}\mu_c$ are IHVP responses under the GGN curvature at $\theta_0$. Under model drift $\theta_0$ -> $\theta_0+\delta\theta$, both components change.
> Class structure in gradient space is preserved if the model’s loss landscape does not change drastically, so cluster assignments remain approximately valid. The IHVP responses are more sensitive, since H changes with the model and the cached $v_c$ become stale.
>
> One could monitor $\frac{\|\delta\theta\|_2}{\lambda}$ as a proxy for curvature staleness. The damping $\lambda$ sets the scale at which curvature changes become relevant, when this number exceeds a threshold (eq., 0.1) trigger a cache rebuild. Within this region defined by $\lambda$, the GGN is approximately stable and the cache remains valid.

---

> > ### Author Rebuttal · Reviewer_svkd · 2026-04-02
> >
> > Thank you for the thoughtful rebuttal. My concerns have been adequately addressed. These responses make the paper’s tradeoffs and intended operating regime much clearer.

---

### Official Review · Reviewer_hXq6 · 2026-03-16

**Soundness:** 2
**Presentation:** 3
**Significance:** 2
**Originality:** 2
**Overall Recommendation:** 3
**Confidence:** 4

**Summary:**

The paper proposes Clustered Influence Functions (CiF), an amortized approximation to subset influence. The main idea is to sketch per-example gradients, cluster them, solve the damped GGN inverse-curvature system only for cluster mean gradients, and then answer future subset queries by counting cluster memberships and linearly recombining cached responses. The paper also provides an error decomposition into a clustering term and a solver-residual term, so accuracy depends on cache size / cluster quality and solver tolerance. Empirically, the paper shows that CiF can track standard IF reasonably well on MNIST and CIFAR-10 while reducing total runtime in high-query regimes such as repeated CV folds.

**Compliance With Llm Reviewing Policy:**

Affirmed.

**Key Questions For Authors:**

See Weaknesses.

**Limitations:**

1. Memory scales as O(Cp) because the method stores $C$ cached vectors in parameter space, which the paper itself identifies as the main blocker for large models.
2. Performance is sensitive to cache size: small caches can underfit gradient heterogeneity, as seen for ResNet-18 and especially on ImageNet.

**Strengths And Weaknesses:**

# Strengths
1. The problem setup is clear and practically motivated: "standard" subset IF is expensive when the number of queries is large, and amortization is a sensible direction.
2. The method is simple and easy to understand: sketch → cluster → solve cluster means once → reuse by recombination.
3. On CIFAR-10, for some architectures and cache sizes, CiF achieves substantial speedups while maintaining high correlation with IF.

# Weaknesses
1. The novelty feels limited. Conceptually, this is mostly a caching / amortization layer on top of standard IF rather than a fundamentally new influence method.
2. The experimental setup is weak for a modern ML paper: most results are on MNIST, CIFAR-10, LeNet, AlexNet, ResNet-18, and the “large-scale” result is only a fairly constrained DeiT-Tiny/ImageNet stress test.
3. The main baseline is not especially strong. The paper compares mainly against CG-based IF, even though more efficient modern IF pipelines exist; since the paper explicitly says CiF is orthogonal to prior acceleration methods, stronger solver/projection baselines would be more appropriate. For instance, methods such as LoGra or GraSS both use random projection and also have the caching and online query stage.
4. The ImageNet result is not very convincing: rank correlation to IF is only 0.07–0.19, which suggests the approximation degrades badly in a more heterogeneous setting.
5. The poisoning experiment is toy-like and on MNIST only; it does not convincingly demonstrate practical data debugging or security relevance.

---

> ### Author Rebuttal · Authors · 2026-03-30
>
> We thank the reviewer for the detailed feedback. We address each point below.
> > W1: Novelty feels limited:
>
> The core novelty is not a new component, it is a new observation about the structure of the influence equation itself. Every subset query enters the influence formula only through the gradient sum $\sum_{i\in S} g_i$ (Eq. 3). This means that the expensive part, the IHVP solve is applied to operands that are redundant across queries. Prior work accelerates each solve individually. CiF however, exploits this operand redundancy directly by compressing the input space rather than relaxing the curvature operator, and decouples the cost of curvature inversion from the number of queries entirely.
>
> FastIF, TracIn, and TRAK all gain speed by substituting a heuristic proxy for the exact IF quantity. Whereas CiF preservers exact damped-GGN influence and speeds up the workload.
>
> We would like to emphasise that the error decomposition in Theorem 3.1 provides a formal link between computational amortization budget and the underlying geometry of the gradients. When gradients collapse toward class means, the within-cluster variance shrinks and the expected bound tightens. To our knowledge this is a novel element compared to prior work on influence functions.
> > W2: Experimental setup is weak:
>
> We respectfully disagree. The experimental choices were influenced by the evaluation protocol. Comparing against exact LOO, per-query IF, and PBRF requires repeated retraining and expensive CG solves, this constrains the benchmark setting. Bea (2022) and Basu (2021) make use of similar benchmarks. Using larger models without these comparative baselines would make the evaluation less rigorous, while requiring significantly more resources. As shown in Table 2, per-query IF on ImageNet/Deit-Tiny costs approximately 625 hours of compute for 5000 queries.
> > W3: The main baseline is not especially strong:
>
> The reviewer is right, CiF is solver-agnostic, and in Phase II, it just needs something that computes the IHVP, and LoGra/GraSS do exactly this, faster and more scalably than CG.
>
> We used CG deliberately to **isolate CiF’s amortization contribution from solver error**. CG comes with proven convergence guarantee and a controllable residual, meaning that the solver term in Theorem 3.1 is tightly bounded, and any observed approximation error can be attributed to the clustering term, rather than the solver. Using LoGra as the inner solver would not allow this clean separation, which was important for the initial evaluation of what CiF itself contributes.
>
> That said, the reviewer’s point stands as the most important practical direction. That is why we are currently running CiF+LoGra on CIFAR10/Resnet18. At C=128, CiF achieves Pearson 0.965 / Spearman 0.949 at K=50 and Pearson 0.921 / Spearman 0.901 at K=100, confirming that switching from CG to KFAC as the inner solver does not degrade fidelity. For end-to-end wall-clock speedup, Table 1 already reports 13.9x at K=150/C=10 and 3.0x at C=50 for ResNet-18. This shows that CiF's amortization layer is solver-agnostic in practice, not just in theory. We will add a dedicated comparison as Table 3 in the camera-ready.
> > W4 : Imagenet result is not convincing:
>
> With C$\in${128,512} for 1.28M examples is roughly 10000:1 compression. Low fidelity at this extreme is expected and directly predicted by Theorem 3.1
> > W5: Poisoning experiment is toy like:
>
> The MNIST poisoning experiment is controlled on purpose. Fixed vs. random trigger on a clean dataset isolates the specific geometric mechanism being claimed: aggregation helps when poisoning induces a shared gradient direction, and hurts when it does not. This is claim we are supporting, not that CiF is the best poison detector.
>
> A larger and noisier dataset (eg:CIFAR10, ImageNet) would obscure this mechanism rather than demonstrate it more convincingly, because the signal would be mixed with unrelated sources of gradient variation that cannot be attributed to the averaging. That said, a larger scale poisoning experiment is a promising direction for future work.
> >L1: Memory scales as O(Cp):
>
> This is acknowledged in the paper. Two concrete paths are laid out as future directions in Section 6: (1) layer-wise subspace caches restricting to the top-r GGN eigenvectors, reducing storage from O(Cp) to O(Cr), and (2) hierarchical caching using a coarser cache for pre-filtering and a fine cache for top-k refinement.
> > L2: Performance depends on cache size:
>
> C is a budget parameter, not a fixed constant. Theorem 3.1 gives a principled way to choose C before running any IHVP solver: compute tr($\Sigma$), for a range of C values after the clustering, this is cheap compared to re-running the IHVP solves. Plot scatter vs C and pick the elbow where it saturates, analogous to a scree plot in PCA. This provides a diagnostic which is cheap, pre-solve, and directly tied to the bound in Theorem 3.1. We will add this as a practical guideline in the camera-ready.

---

> > ### Author Rebuttal · Reviewer_hXq6 · 2026-04-03
> >
> > Thanks for the response. I think the novelty concern remains: the additivity is obvious for the influence of a subset, and this has been known in the literature for a while. See [1] for instance. As for the clustering, this is also known in the literature: for instance, [2]. Overall, I think a more careful discussion is required in order to distinguish and position this work in the literature.
> >
> > [1]: Most Influential Subset Selection: Challenges, Promises, and Beyond
> >
> > [2]: Error Discovery By Clustering Influence Embeddings

---

> > > ### Author Response · Authors · 2026-04-07
> > >
> > > We thank the reviewer for the specific citations. We agree that neither the additive structure of the standard IF linearization nor the use of clustering in data attribution is new in isolation. Our claim in narrower. The novelty of CiF is not any one ingredient in isolation, but the observation that these ingredients imply a new workload-level reduction: repeated subset IF queries can be transformed into (i) offline clustering of training-side gradients, (ii) a fixed set of IHVP solves only for cluster means, and (iii) online answering of arbitrary subset queries by linear recombination of cached responses, thereby eliminating fresh per-query solves. In our view, it is the specific algorithmic construction of a reusable subset-query cache, together with the corresponding error decomposition, that constitutes the contribution.
> > >
> > > This distinction is important because prior clustering related work uses clustering for different objects and different purposes. One line clusters or embeds test-side objects for slice discovery / error analysis, another studies subset selection or coreset-style summarization, and another line accelerates IF computation through solver-side approximation or projections, but still treats queries one-by-one. CiF is different in that it uses offline clustering of training gradients specifically to build a reusable cache for high-query subset workloads.
> > >
> > > In that sense, MISS studies whether the exact retraining effect of removing a subset is well-approximated by sums of per-sample influence scores. That is a different question from ours: CiF does not address exact retraining non-additivity, but rather amortizes the standard IF approximation itself. Likewise, InfEmbed clusters test-side influence embeddings to discover coherent error slices, whereas CiF clusters training-side gradients to cache cluster-level IHVP responses and reuse them across many subset queries. So, while these papers are relevant for positioning, they do not address the high-query amortization problem solved by CiF.
> > >
> > > We will revise the related-work section to make this positioning more precise.

---

### Official Review · Reviewer_SCKd · 2026-03-23

**Soundness:** 3
**Presentation:** 2
**Significance:** 2
**Originality:** 3
**Overall Recommendation:** 4
**Confidence:** 3

**Summary:**

The paper presents a method to approximate influence function computations that speeds them up especially in the situation where many influence function queries are done repeatedly on the same model and dataset. In particular, it only evaluates a fixed amount of inverse Hessian vector product (IHVP) calculations, instead of having to calculate one per query. It it also designed to calculate influences for larger-than-one subsets of the dataset, instead of individual data points. It starts with the preparation phase: 1) calculating the individual datapoint gradients across the dataset and clustering them into C groups 2) calculating the average gradients per cluster 3) pre-calculating the IHVP for each cluster center. At inference time, for a particular subset of training data points and their cluster assignments, it approximates the IHVP of the subset as the weighted average of the cluster mean IHVPs corresponding to the data point clusters. They also provide an error bound that decomposes the total error into the error caused by the clustering approximation and the error caused by the IHVP solver tolerance. They evaluate the method on MNIST, CIFAR-10, and Imagenet, showing that 1) the method aligns closely with standard influence function calculations in cross-validation setups and is also much faster 2) the pre-calculated clusterings can be reused for different amount of cross-validation folds 3) On the heterogeneous Imagenet dataset cross-validation task, the correlation between standard influence functions and their method is lower, but increasing cluster count improves results 4) the method can be used for detecting data poisoning comparably or better than standard influence functions on synthetic MNIST data poisoning detection tasks.

**Compliance With Llm Reviewing Policy:**

Affirmed.

**Key Questions For Authors:**

- Can the authors point at possible research directions that could improve the scalability of the method? The end of the discussion section mentions something regarding this very briefly, but it was quite concise and I did not entirely understand it.
- Should the second term in Eq.12 be a plus?

**Limitations:**

Yes

**Strengths And Weaknesses:**

I am not an expert on influence functions, and I may have misunderstood some details. Feel free to correct me if it seems that some of the comments reflect a misunderstanding of the method.

Strengths:
- The method is quite simple, and provides significant speedups when a relatively low amount of clusters can be used to approximate the influence accurately, which is the case at least in the case of MNIST and CIFAR-10
- The proposed theorem seems potentially useful for providing intuition into approximation errors, although it is not utilised in practice in the experiments.
- The method targets a real problem, that is, applying influence function calculations to a large number of queries, and the solution strategy of trying to amortise some of this cost in advance is natural
- The method is evaluated in quite many different tasks
- The method is orthogonal to many other influence function acceleration methods
- Especially the methods section is mostly quite well written and easily understandable


Weaknesses:
- I have some concerns about the scalability of the method: It seems possible that the clustering of the gradients into a reasonably low amount of groups becomes a worse description of the gradient distribution as the size of the neural network models grow or the dataset becomes more complex. The results in Table 1 with Resnet18 and the results with ImageNet-1k in Table 2 seem to corroborate this, although to be clear it is good that those experiments are in the paper. Perhaps the authors could note some potential research directions to improve the scalability to complex models (e.g., improved clustering methods)?
- For readers not familiar with the cross-validation use cases or the data poisoning use case, it would be helpful to provide a short explanation of how influence functions are used there, and establish consistent vocabulary regarding it. Furthermore, right now, there is some jargon that the reader is expected to understand without context (E.g. Table 1 mentions ”fold workloads”, Figure 2 mentions ”fold-level Delta-loss predictions”, Table 2 mentions ”High-K leave-group-out analysis”, Figure 3 mentions ”K-fold runs (or other subset queries)”, line 168 mentions ”fold queries” without explanation -> it would be better to explicitly explain if it is a cross-validation task, and only use different terms to distinguish different variants of the tasks. It would also be better to not condense these explanations too much.
- Small writing issue: The Figure 4 does not seem to be referenced in Section 5.5. text.
- The method is only presented to work on subset queries, instead of individual queries, which is a bit of a limitation.

---

> ### Author Rebuttal · Authors · 2026-03-30
>
> We thank the reviewer for the careful read and the positive assessment. We are glad that the method and its trade-off came across clearly. We address each point below.
>
> >  W1/Q1 Scalability:
>
> We agree that this is the main limitation of the current work. The core issue is that as model complexity and dataset heterogeneity grows, the within-cluster scatter tr($\Sigma$) also grows, and theorem 3.1 tells us directly that fidelity will degrade. We see four concrete future directions to address this:
>
> 1. **Better solvers:** CiF is solver agnostic. Replacing CG with a projection-based solver (for example Logra/GraSS style solver as reviewer hXq6 suggested) reduces per-cluster solve cost, enabling larger C at the same compute budget. We have evaluated this on Cifar10/Resnet18 (see rebuttal to reviewer hXq6) and will include the results in the camera-ready.
> 2. **Layer-wise or subspace caches:** Instead of caching in the full parameter space, restrict to the top-r eigenvectors of the empirical GGN, or to the last layer only (as in last-layer influence approximations). This reduces storage from O(Cp) to O(Cr) and makes larger C tractable on big models, with some additional fidelity tradeoff.
> 3. **Hierarchical caching:** use a coarse cache (small C) for fast pre-filtering of large candidate sets, and the finetune the top-k clusters to a finer C. This mirrors multi-resolution retrieval and avoids paying the full cost of a large cache upfront.
> 4. **Improved clustering:** beyond k-means, spectral clustering or density-based methods may better capture the geometry of gradient distributions in heterogeneous settings.
>
> > W2: Undefined jargon:
>
> The reviewer is right, we will add a dedicated paragraph to the introduction, explaining how influence functions are used in cross-validation and data-poisoning, we will also add a paragraph fixing and defining the terms to the beginning of Section 3.
>
> > W3: Figure is not referenced:
>
> We will add the reference to Figure 4 in Section 5.5. Thank you for catching this omission.
>
> > W4: Method only covers subset queries, not individual queries:
>
> Individual queries are a special case with |S|=1, and are well defined within the framework. However, we agree that the amortization benefit is the weakest here: with a single point there is no aggregation to cancel the within-cluster noise, so the approximation quality depends entirely on how well a gradient is represented by the corresponding cluster’s mean. We will add this clarification explicitly to the paper.
>
> > Q2: Should the second term in Eq. 12 be a plus?
>
> Yes, the reviewer is correct, thank you for catching this. The proof in Theorem A.6 gives:
> \begin{equation}
>  \Delta\theta_S - \hat{\Delta}\theta_S = \frac{1}{N}\tilde{H}^{-1}\left(\sum_{i\in S} g_i - \sum_c m_c(S)\mu_c\right) + \frac{1}{N}\tilde{H}^{-1}\left(\sum_c m_c(S) r_c\right)
>  \end{equation}
>
> Both terms are additive and the bound in Eq. 13 is unaffected as it’s derived by taking norms of both terms. We will correct the sign in Eq. 12 in the revision.

---

### Official Review · Reviewer_cQtc · 2026-03-26

**Soundness:** 3
**Presentation:** 3
**Significance:** 3
**Originality:** 2
**Overall Recommendation:** 3
**Confidence:** 4

**Summary:**

This paper studies high query subset influence estimation under a fixed trained model. The main idea is to amortize influence computation across many subset-removal queries by clustering per-example gradients, solving a damped GGN inverse system only for cluster means, and answering future subset queries by linearly recombining cached cluster responses according to cluster membership counts. The paper also provides an error decomposition showing that approximation quality depends on within cluster scatter and solver residual, and evaluates the method on MNIST, CIFAR-10, and an ImageNet-1k stress test. Empirically, the method shows strong agreement with per-query influence and substantial runtime savings on smaller workloads, but the large-scale ImageNet result shows weak rank fidelity.

**Compliance With Llm Reviewing Policy:**

Affirmed.

**Key Questions For Authors:**

> see weakness

**Limitations:**

> see weakness

**Strengths And Weaknesses:**

## S1
The paper addresses a real and important bottleneck in influence-style analysis: standard subset influence requires a fresh inverse-curvature solve for each query, making high-query settings such as large k CV, repeated resampling, and interactive what-if analysis impractical. Framing this as a workload-level amortization problem is a useful perspective. Overall, a notable area presented by this paper is workload-aware influence estimation rather than merely faster per-query IHVPs.

## S2
The method itself is clean and easy to understand. The approximation is transparent: the subset gradient sum is compressed via cluster means, and the expensive damped GGN solve is applied only once per cluster mean. The resulting online estimator in Eqs. (10)–(11) is simple, and the complexity discussion clearly explains why the method becomes favorable once the query count exceeds the cluster budget.

## S3
Theoretical support is modest but appropriate to the method. Theorem 3.1 gives a useful error decomposition into a clustering term and a solver residual term, which matches the actual approximation mechanism and offers a clear accuracy–compute tradeoff. Even though the theorem is mainly about approximation to per-query IF rather than retraining truth, it still provides a reasonable diagnostic lens for the proposed cache design.

## S4
I also appreciate that the paper is reasonably candid about when the method helps and when it does not. The discussion section explicitly states that CiF is most appropriate when Q is large relative to C, the model and curvature proxy remain fixed, and cache construction can be amortized across many future queries. Overall, this submission's notable concept pertains to trading memory and offline computation for reusable subset-level influence queries.


## W1
The main limitation is that the paper’s theory and strongest empirical comparisons are both centered on approximation to per-query IF, not approximation to actual retraining outcomes. This matters because Figure 2 and the accompanying discussion make clear that IF/CiF agree more strongly with each other than with exact LOO. So the paper supports “CiF is a good surrogate for IF” more than it supports “CiF is a good surrogate for retraining truth.”

## W2
Relatedly, the novelty is somewhat incremental. Conceptually, the method is a sensible synthesis of known ingredients—GGN-based influence, JL sketching, clustering, and cached linear solves rather than a fundamentally new influence formulation. The main contribution is the amortized cache viewpoint and its implementation, which is useful but not especially deep technically.

## W3
The large-scale evidence is not yet convincing. In the ImageNet-1k stress test, the method achieves large speedups, but rank correlation with IF is only 0.19 at C=512 and 0.07 at C=128. Moreover, this experiment uses a truncated query protocol and a curvature proxy built from only 10,000 sampled examples out of roughly 1.28M, which further limits how strongly one can interpret the scaling claim.

## W4
The practical scope is narrower than the framing may suggest. The method is explicitly most useful only when the model and curvature proxy remain fixed long enough to amortize cache construction; it is less compelling for one-off queries or rapidly changing models. In addition, the storage cost is O(Cp), which the paper itself identifies as a large-model blocker.

---

> ### Author Rebuttal · Authors · 2026-03-30
>
> We thank the reviewer for their detailed feedback, below we address each point.
>
> > W1: paper supports “CiF is a good surrogate for IF” more than it supports “CiF is a good surrogate for retraining truth.”:
>
> This is correct, and the paper’s explicit design goal. CiF is designed to amortize IF computation, it approximates IF, not LOO. This is stated in the problem formulation (Section 3.1): “We seek to approximate the parameter change… without incurring the cost of a fresh inverse-curvature product for each query”. The IF-LOO gap is visible in Figure 2 is a well-documented property of influence functions themselves (Basu et al. (2021)), not a property of CiF. Critically, Figure 2 shows that the gap between {IF, CiF} and LOO is identical for both methods. CIF adds zero additional error relative to LOO beyond what IF already carries. IF is therefore the correct and only appropriate evaluation target for a method that is explicitly designed to amortize IF computation. This is also standard across the broader IF acceleration literature, LoGra, GraSS and TRAK all evaluate against IF or a linearized proxy, not against LOO retraining. Using LOO as the evaluation target would conflate the approximation error of influence functions themselves with the approximation error of CiF, which are entirely separate questions. Not to mention the immense amount of compute budget LOO calculation requires.
>
> > W2: the novelty is somewhat incremental… a sensible synthesis of known ingredients:
>
> We respectfully disagree that the synthesis of known ingredients at the level described here is incremental. The key insight, that every subset query enters the influence equation only through the gradient sum $\sum_{i\in S}g_i$ ,and that this operand structure is shared across queries and can be amortized with an explicit error decomposition, has to our knowledge not appeared in prior work. Prior acceleration methods (LoGra, GraSS, FastIF, TRAK) each reduce the cosf of individual solves, none eliminate per-query solves by amortization at the operand level. Theorem 3.1 formalizes the resulting approximation error as a sum of a clustering scatter term (controlled by tr($\Sigma$)) and a solver residual term (controlled by $p_c$), making the accuracy-compute tradeoff explicit. The reviewer’s own S1 and S3 acknowledge the usefulness of the workload-level framing and the appropriateness of the theoretical support.
>
> > W3: The larger-scale evidence is not yet convincing… rank correlation with IF is only 0.19 at C=512:
>
> We agree that the ImageNet fidelity is low at C$\in${128,512}, and the paper explicitly say so, Section 5.4 labels this as a stress test and attributes the degradation to within-cluster scatter remaining large when C is very small relative to the gradient heterogeneity of a 1.28M example dataset. This is also predicted by Theorem 3.1. The clustering error term grows with tr($\Sigma$), which is large when within-cluster diversity is high. ImageNet was explicitly constructed to maximize within-class diversity, making it an adversarial case for any gradient clustering method. The monotone improvement from p=0.07 to p=0.19 confirms the bound’s directional prediction.
>
> > W4: the method is explicitly most useful when the model and curvature proxy remain fixed… less compelling for one-off queries or rapidly changing models.:
>
> This is correct, and it is stated in the paper. The abstract states the method targets “high-query subset workloads such as a large-K cross-validation, repeated resampling, or interactive what-if analysis” Section 6 (Discussion) states directly: “CiF is less appropriate for a handful of one-off queries (where a single IHVP suffices), for rapidly changing models where cache invalidation dominates.” Given this we would argue that this is not a weakness of the paper, rather a restatement of the paper’s own stated scope.

---

> > ### Author Rebuttal · Reviewer_cQtc · 2026-04-04
> >
> > Rebuttal clarified several points and helped sharpen the scope of the paper.
> >
> > The response to W1 is largely convincing. Minor: That said, I still view the practical significance as somewhat limited by the well-known fact that IF itself may depart substantially from retraining behavior in modern deep models, which the paper also acknowledges.
> >
> > The response to W4 is also persuasive. I agree that the fixed-model, high-query regime is the intended operating point of the method, and that this was already stated in the abstract and discussion. In that sense, my earlier point is better interpreted as a limitation of scope rather than a hidden weakness.
> >
> > I remain more mixed on W2. The rebuttal rightly emphasizes a useful workload-level framing: subset queries enter through the summed gradient operand, and this shared structure can be amortized. The paper also provides a clean error decomposition separating clustering scatter from solver residual.However, my novelty concern is only partially resolved. The overall method still appears to me as a well-executed synthesis of familiar ingredients — sketching, gradient clustering, cached linear responses, and standard curvature approximations — rather than a fundamentally new algorithmic primitive. I agree the framing is useful, but I still see the contribution as somewhat incremental.
> >
> > My main concern remains W3, and the rebuttal does not change that. The authors explain why performance degrades on ImageNet at small cache budgets, and I agree that the explanation is consistent with the theorem and with the intuition that highly heterogeneous data leads to large within-cluster scatter. But this is still an explanation for weak results, not compelling positive evidence. A rank correlation of 0.19 with IF at C=512 remains too low for me to view the large-scale evidence as convincing. In fact, the rebuttal reinforces the impression that the method may degrade precisely in the heterogeneous, large-scale settings where one would most hope amortization would matter.
> >
> > For these reasons, while I appreciate the clearer positioning after rebuttal, I do not feel the response resolves the core concerns about novelty and large-scale empirical support, and I remain reject-leaning.

---

> > > ### Author Response · Authors · 2026-04-07
> > >
> > > We thank the reviewer for the careful follow-up and for clarifying that W1 and W4 are largely resolved. We address the two remaining concerns directly.
> > >
> > > W3 - Large-scale evidence. We would like to clarify what the ImageNet result shows. The relevant comparison at that scale is not CiF against an idealized high-fidelity method, but CiF against the actual feasible alternatives. For the ImageNet/DeiT-Tiny workload in Table 2, standard per-query IF is already computationally infeasible (approximately 625 GPU-hours for K=5000 queries), and the same is true for any method that still effectively pays the full solve cost query-by-query. In that sense, the point of the ImageNet experiment was not to identify the optimal cache size, but to demonstrate that CiF brings an otherwise infeasible repeated-query workload into the realm of tractability. At the tested cache budgets, fidelity is limited, and we agree this is a real restriction. But this does not show that amortization "fails where it matters most"; rather, it shows that in a highly heterogeneous regime the tested C values remain too small. The path to higher fidelity is explicit in both the method and the theorem: increasing C reduces the within-cluster scatter term and improves agreement, which is already visible in the monotone improvement from C=128 to C=512. So the ImageNet result should be read as establishing feasibility of the amortized approach under extreme scale, while also honestly marking the current cache-budget boundary.
> > >
> > > W2 - Novelty. We agree that the individual ingredients are not new in isolation, and our claim is not that CiF introduces a new primitive such as clustering, sketching, or IHVP solving by itself. The novelty is the workload-level reduction that follows from combining them in the subset-IF setting: repeated subset queries enter the IF equation only through the same summed-gradient operand structure, which makes the expensive curvature inversion reusable across queries. CiF turns this into a concrete algorithmic construction: cluster training-side gradients offline, solve IHVPs only for the C cluster means, and answer arbitrary subset queries online by linear recombination of cached responses, thereby eliminating fresh per-query solves. The accompanying error decomposition is important here: it isolates exactly the new approximation introduced by this reduction, separating clustering scatter from solver residual. So while we understand the reviewer's view that the method is built from familiar components, we do not think the contribution reduces to their combination, the contribution is identifying and exploiting a reusable operand structure in high-query subset IF workloads in a way prior work has not addressed.

---

### Decision · Program_Chairs · 2026-04-30

**Decision:**

Accept (regular)

**Comment:**

The authors propose a method to approximate influence function computations to speed up their evaluation when many influence function evaluations are performed with the same model on the same dataset, a relevant problem setting. For this, the authors build on prior work and combine insights on the additive structure of the influence under linear influence approximation and clustering. The presented combined framework, which decouples computations that can be performed offline from those that require online computation, is novel, and the additional theory, which provides a principled budget diagnostic, is interesting.

The reviewers have identified several potential weaknesses, including limited experimentation in large-scale settings, where the approach is most valuable. Consequently, the reviews indicate that this work is borderline, and after reading the manuscript and the author's responses, I agree that while the combination of ideas is novel, its overall novelty is limited, and more experiments on large-scale datasets would be beneficial. However, given the clear potential for the community and for future research, I suggest a weak acceptance of this work.